# Enhanced NF-κB signaling in type-2 dendritic cells at baseline predicts non-response to adalimumab in psoriasis

Rosa Andres-Ejarque [1], Hira Bahadur Ale[1], Katarzyna Grys[1,2], Isabella Tosi[1,2], Shane Solanky[1],
Chrysanthi Ainali[1,3], Zeynep Catak[1,2], Hemawtee Sreeneebus[1,2], Jake Saklatvala [4], Nick Dand [4],
Emanuele de Rinaldis[2,12], Anna Chapman[5], Frank O. Nestle[1,2,12], Michael R. Barnes [6], Richard B. Warren[7,13],
Nick J. Reynolds[8,13], Christopher E. M. Griffiths[7,13], Jonathan N. Barker[1,2,13], Catherine H. Smith[1,2,13],
Paola Di Meglio [1,2 ✉] & the PSORT Consortium*

Biologic therapies have transformed the management of psoriasis, but clinical outcome is variable leaving an unmet clinical need for predictive biomarkers of response. Here we perform in-depth immunomonitoring of blood immune cells of 67 patients with psoriasis, before and during therapy with the anti-TNF drug adalimumab, to identify immune mediators of clinical response and evaluate their predictive value. Enhanced NF-κBp65 phosphorylation, induced by TNF and LPS in type-2 dendritic cells (DC) before therapy, significantly correlates with lack of clinical response after 12 weeks of treatment. The heightened NF-κB activation is linked to increased DC maturation in vitro and frequency of IL-17+ T cells in the blood of non-responders before therapy. Moreover, lesional skin of non-responders contains higher numbers of dermal DC expressing the maturation marker CD83 and producing IL-23, and increased numbers of IL-17+ T cells. Finally, we identify and clinically validate LPS-induced NF-κBp65 phosphorylation before therapy as a predictive biomarker of non-response to adalimumab, with 100% sensitivity and 90.1% specificity in an independent cohort. Our study uncovers important molecular and cellular mediators underpinning adalimumab mechanisms of action in psoriasis and we propose a blood biomarker for predicting clinical outcome.

[1] St. Johns Institute of Dermatology, Kings College London, London, United Kingdom. [2] NIHR Biomedical Research Centre, Guys & St Thomas NHS Foundation Trust & King's College London, London, UK. [3] DIGNOSIS Limited, London, UK. [4] Department of Medical and Molecular Genetics, Kings College London, London, UK. [5] Dermatology Department, Queen Elizabeth Hospital, Lewisham and Greenwich NHS Trust, London, UK. [6] Centre for Translational Bioinformatics, William Harvey Research Institute, Queen Mary University of London, Charterhouse Square, London, UK. [7] Dermatology Centre, Salford Royal Hospital, The University of Manchester, Manchester Academic Health Science Centre, Manchester, UK. [8] Institute of Translational and Clinical Medicine, Newcastle University Medical School and Department of Dermatology, Royal Victoria Infirmary, Newcastle Hospitals NHS Foundation Trust, Newcastle upon Tyne, UK. [12] Present address: Sanofi, Cambridge, MA, USA. [13] These authors contributed equally: Richard B. Warren, Nick J. Reynolds, Christopher E. M. Griffiths, Jonathan N. Barker, Catherine H. Smith. *A list of authors and their affiliations appears at the end of the paper. ✉email: paola.dimeglio@kcl.ac.uk

Psoriasis is a common, chronic, immune-mediated, inflammatory skin disease, affecting over 100 million people worldwide and is recognized as a serious non-communicable disease by the World Health Organization for the significant negative impact it has on people's lives[1].

Painful, disfiguring and disabling skin lesions result from the combination of genetic susceptibility, environmental triggers and dysregulated immune responses involving dendritic cells (DC), IL-17-producing T (IL17[+] T) cells, keratinocytes, and pro-inflammatory cytokines[2,3]. Over the last two decades, significant advances in understanding the pathogenic mechanisms underpinning psoriasis, have led to the adoption of biological treatments ("biologics") targeting the key cytokines TNF, IL-23 and IL-17A, which have transformed disease management. Nevertheless, response to biologics is often heterogeneous, with lack or loss of response being a persistent issue in up to 30% of patients, in addition to potential side-effect, such as infections and allergic reactions[4]. Variability of response and treatment failure results in treatment switching, which in turn increases both patients' dissatisfaction and costs for healthcare providers. Thus, psoriasis could greatly profit from the implementation of stratified medicine approaches to benefit patients and reduce costs[5]. The choice of which biologic to prescribe is currently based on clinical factors, for example, anti-TNF are preferentially prescribed to patients with concomitant psoriatic arthritis[6]. The PSORT consortium, a partnership involving clinicians, translational researchers, industry and patient groups, aims to identify determinants of response to biologic therapies, and ultimately deliver a patient stratifier to guide psoriasis management[7]. The analysis of demographic, social, clinical, pharmacological and genomic data has identified a number of factors, such as ethnicity, weight, smoking, disease severity[8], serum drug levels[9] and HLA-Cw0602 status[10], which are associated with outcome to biologics. However, the small effect size suggests that these factors alone are insufficient to inform optimal treatment selection. Thus, there is an unmet clinical need to identify biomarkers predictive of response to biologics. In particular, biomarkers involved in disease pathogenesis and/or drug mechanism of action (i.e. mechanistic biomarkers) are considered more informative, robust and actionable than biomarkers simply reflecting the clinical outcome[11].

Anti-TNF inhibitors were the first class of biologics approved for psoriasis, in line with the pleiotropic role of TNF in inflammation. Activation of the transcription factor nuclear factor-κB (NF-κB) is a key event downstream TNF signalling, inducing the expression of inflammatory and anti-apoptotic genes, which influence critical cellular behaviours such as activation, maturation, migration, proliferation and survival[12]. Adalimumab is a recombinant, fully humanized, IgG antibody with high affinity and specificity for both soluble and membrane-bound TNF. In clinical trials, 71% of patients with moderate-to-severe psoriasis achieved a 75% reduction in the standard objective disease severity score used in psoriasis—Psoriasis Area and Severity Index (PASI; PASI75 response) after 16 weeks of adalimumab treatment, with 45% achieving 90% reduction (PASI90 response)[13]. Whilst newer biologics have been developed and licensed, adalimumab remains a first-line intervention in psoriasis, given its effectiveness and well-established safety profile and —with the advent of biosimilars—significantly reduced costs. Nevertheless, the cellular and molecular mechanisms underpinning its clinical efficacy are ill understood. In a small early study, adalimumab induced normalization of keratinocyte differentiation and decreased the number of T cells and DC in psoriasis skin lesions[14], in keeping with the down-modulation of myeloid and Th17-response genes observed in patients treated with etanercept[15,16]. Little is known about the effect of adalimumab on circulating blood immune cells in psoriasis. In particular, the effect of TNF-neutralization on proximal signalling events, such as NF-κB activation, has not been investigated thus far.

Here, as a fundamental step to identify mechanistic immune mediators of response to adalimumab and evaluate their value as predictive biomarkers, we investigate the blood cellular target(s) and molecular effect of adalimumab in psoriasis. We utilise the clinical psoriasis bioresource of the PSORT Consortium[7] and perform in-depth functional and phenotypic immunomonitoring of blood immune cells of 67 patients with psoriasis and 20 healthy controls, before and during biologic therapy. We validate our findings with mechanistic in vitro experiments and skin in situ imaging studies. Finally, we compare the predictive value of the most promising immune traits identified and validate an immune biomarker predictive of response in an independent patient cohort.

## Results

**Adalimumab blocks NF-κB translocation in lymphoid cells.** To obtain insights into the cellular target and effect of adalimumab in blood we monitored NF-κB activation in 16 patients undergoing adalimumab treatment (PSORT adalimumab discovery cohort, Supplementary Fig. 1, Supplementary Data 1), measuring NF-κB nuclear translocation in seven major immune cell populations by imaging flow cytometry. Fresh whole blood, obtained at baseline before starting therapy (week 0, w0) and at week 1 (w1), 4 (w4) and 12 (w12) after the initiation of the treatment, was stimulated with either TNF or LPS (as additional well-known NF-κB-inducer) or left unstimulated, and NF-κB nuclear translocation was quantified as Rd score (Eq. (1) see methods) in neutrophils, monocytes, DC, natural killer (NK), NKT, CD4[+] and CD8[+] T cells (Fig. 1a and Supplementary Fig. 2). Psoriasis patients receiving the anti-12/23p40 mAb ustekinumab ($n = 24$) and matched healthy volunteers ($n = 11$) were used as control groups (Supplementary Fig. 1). No difference in NF-κB constitutive nuclear localization, that is, in absence of stimulation, was observed at any time point in any cell type (Supplementary Fig. 3a). Following stimulation, TNF induced NF-κB translocation in both lymphoid and myeloid cells, while, as expected, LPS only activated neutrophils, monocytes and DC in blood collected at w0 (Fig. 1b, c). A similar activation profile was observed in the whole blood of healthy volunteers (Supplementary Fig. 3b). NF-κB activation induced by TNF in lymphoid cells was strongly inhibited in the blood of patients receiving adalimumab at w1, w4 and w12 of treatment (Fig. 1b, c), but not in those receiving ustekinumab (Supplementary Fig. 3c), in line with the molecular pathway targeted by each drug. Adalimumab exerted the strongest inhibitory effect in lymphoid cells, i.e., CD4[+] T, CD8[+] T, NK and NKT cells, where NF-κB translocation was significantly inhibited by 70–96% (Fig. 1b,c and Supplementary Data 2). In contrast, adalimumab therapy resulted in a weaker inhibition in myeloid cells, with only partial inhibition in DC (w1, 50%, FDR = 0.0061; w4, 50%, FDR = 0.0066; w12, 70%, FDR = 0.0086), and no effect in monocytes and neutrophils (Fig. 1b, c, and Supplementary Data 2). To confirm this inhibition pattern, we stimulated whole blood obtained from healthy volunteers ($n = 3$) with increasing concentration of TNF, in the presence or absence of adalimumab added to the culture. In keeping with the results obtained in patients undergoing adalimumab therapy, TNF-induced NF-κB nuclear translocation was significantly inhibited by up to 98% in T, NK and NKT cells, but only partially in DC (70.48%, FDR = 0.084) with no effect on neutrophils and monocytes (Supplementary Fig. 3d). As expected, neither adalimumab nor ustekinumab therapy had any effect on LPS-induced

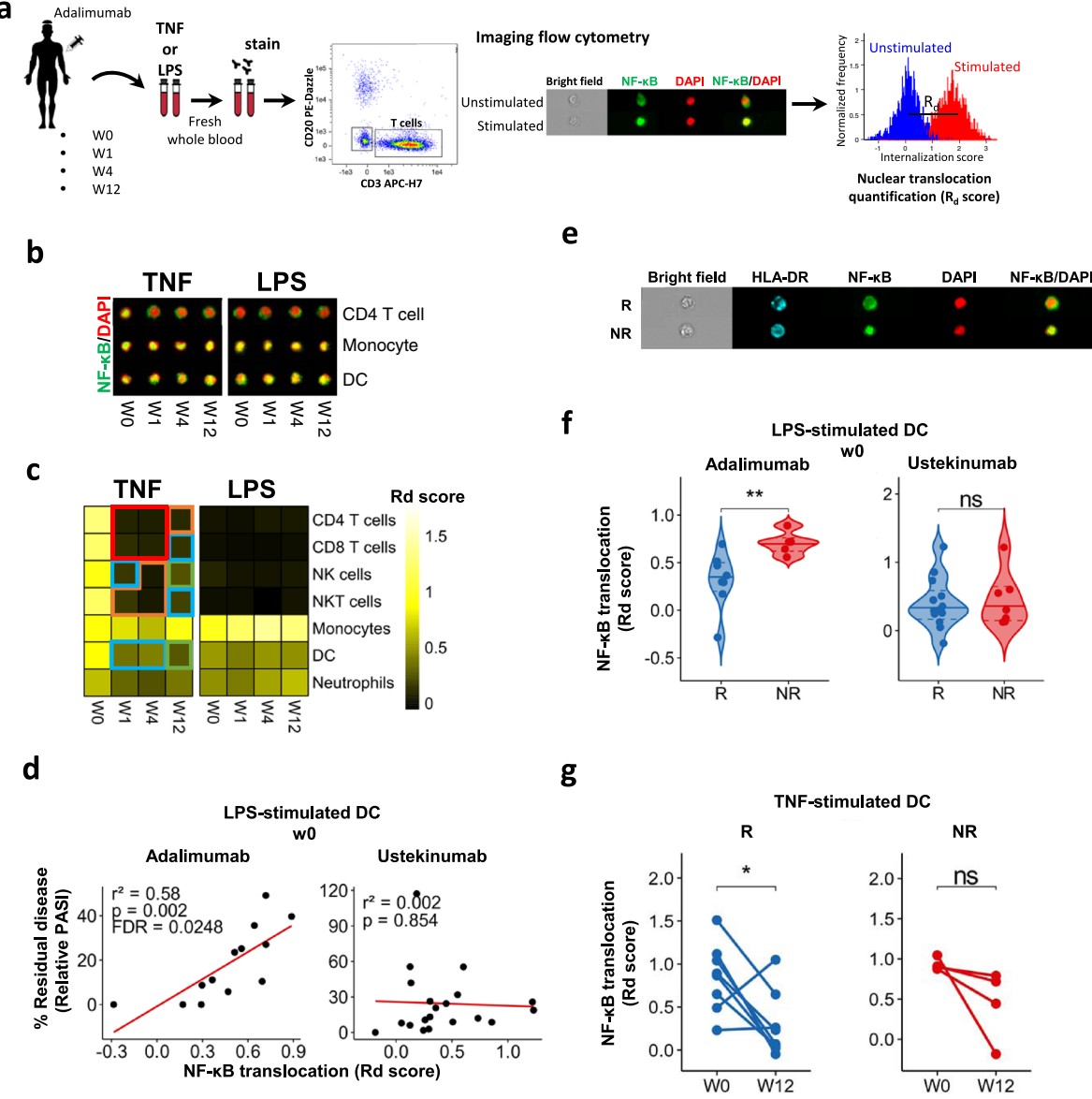

**Fig. 1 LPS-induced NF-κB translocation in DCs at baseline correlates with response to adalimumab. a** Cartoon depicting the experimental workflow. Whole blood from psoriasis patients collected at baseline (week0, w0) and at w1, w4 and w12 after initiation of adalimumab therapy was stimulated with TNF, LPS or left unstimulated, stained with fluorescent antibodies and NF-κB nuclear translocation was measured in distinct cell populations identified by imaging flow cytometry, as shown in the representative images included in the cartoon. **b** Representative fluorescent images of selected cell populations stained for NF-κB (green) and nucleus (DAPI, red); their colocalization is shown in yellow. **c** Heatmap showing NF-κB nuclear translocation measured as Rd score in immune cell populations ($n = 13$–$16$). **d** Correlation analysis between NF-κB translocation (measured as Rd score) at w0 and clinical response expressed as percentage of residual disease at w12 (measured as relative PASI, i.e. PASI at w12/PASI at w0 X 100) in LPS-stimulated dendritic cells (DCs) in PSORT adalimumab ($n = 13$) and ustekinumab cohort ($n = 24$). Each dot represents one patient. **e** Representative fluorescent image of LPS-stimulated DCs at w0 from PASI75 adalimumab responder (R) and non-responder (NR). **f** Violin plot graphs of NF-κB translocation in LPS-stimulated DCs at w0 for PASI75 adalimumab and ustekinumab responders (blue, adalimumab $= 8$; ustekinumab $= 13$) and non-responders (red, adalimumab $= 5$; ustekinumab $= 7$) **$p = 0.0062$. **g** NF-κB translocation in TNF-stimulated DCs at w0 and w12 in adalimumab PASI75 responders ($n = 8$) and non-responders ($n = 4$). Red frame: FDR $< 0.0001$, orange frame: FDR $< 0.001$, blue frame: FDR $< 0.01$, green frame: FDR $< 0.05$ compared to week (w) 0, test. *$p = 0.039$, Wilcoxon test followed by FDR (**c**), Wilcoxon (**g**) or Mann–Whitney U test (**f**). All tests are two-sided. Source data are provided as a Source Data file.

NF-κB translocation in psoriasis patients (Fig. 1b, c; Supplementary Fig. 3c and Supplementary Data 2). Thus, we concluded that the free adalimumab present in the blood stream of patients undergoing therapy completely blocks NF-κB signalling in T, NK and NKT cells but only partially in DC, raising the possibility that the incomplete inhibition observed in DC may affect clinical outcome.

**NF-κB translocation in DCs at baseline correlates with lack of response to adalimumab.** Next, we investigated whether NF-κB nuclear translocation, induced by TNF or LPS, correlated with clinical response to adalimumab. Response was assessed applying either a continuous model, where the outcome is % residual disease at week 12 measured by relative PASI (i.e. PASI at week 12 divided by PASI at w0 × 100), or a binary model, using gold

standard PASI75, as well as PASI90. No correlation between TNF-induced NF-κB nuclear translocation at any time point and % residual disease at w12 was detected in any cell type (Supplementary Data 3), despite the inhibitory effect of adalimumab in lymphoid cells (Fig. 1c). Thus, the complete inhibition of NF-κB signalling in T, NK and NKT cells is not likely to underpin clinical response to adalimumab at w12. However, we found a statistically significant correlation ($r^2 = 0.58$, $p = 0.002$, FDR < 0.05, linear regression $t$ test) between LPS-induced NF-κB translocation at w0 in DCs and % residual disease, (Fig. 1d and Supplementary Data 3). In particular, patients with higher residual disease at w12 displayed increased NF-κB activation at baseline. Importantly, this correlation was not observed in ustekinumab-treated patients (Fig. 1d), indicating that NF-κB activation in DCs is a specific marker of clinical response to adalimumab, and implicates DC NF-κB activation in the cellular and molecular mechanisms underlying adalimumab response. Next, we assessed whether adalimumab serum levels[9] may have influenced the post-w0 results of the correlation analysis. Linear regression of relative PASI on NF-kB translocation and drug concentration simultaneously, confirms that clinical response is not correlated with NF-kB nuclear translocation in any combination of stimulation, cell type and time point, even after accounting for variation in drug concentration between patients (FDR > 0.05, Supplementary Data 4). Moreover, LPS-induced NF-κB translocation at w0 was significantly ($p < 0.01$, Mann–Whitney $U$ test) lower in DCs of patients achieving PASI75 and PASI90 response (i.e responders, (R) than in non-responders (NR) (Fig. 1e, f and Supplementary Fig. 3e), while no difference was observed in the ustekinumab cohort (Fig. 1f). Finally, TNF-induced NF-kB activation in DC at w12 was significantly reduced by adalimumab in responders but only showed a non-statistically significant ($p = 0.12$, Wilcoxon test) downward trend in non-responders (Fig. 1g). Thus, NF-κB activation in DC prior to therapy differs in adalimumab responders and non-responders, and DC in non-responders may be more refractory to the inhibitory effect of adalimumab therapy.

**NF-κB phosphorylation in cDC2 at baseline correlates with lack of response to adalimumab.** The NF-κB activation cascade includes phosphorylation of p65 NF-kB subunit, such as at Ser529, which regulates NF-κB function[17]. Thus, to replicate and refine our findings we developed a 13-colours phospho-flow cytometry panel (Supplementary Fig. 4) to study NF-κB phosphorylation in 11 cell subsets within cryopreserved PBMCs of patients receiving adalimumab. The use of cryopreserved PBMCs allowed us to extend our overall sample size up to 43 patients (PSORT adalimumab combined cohort, Supplementary Fig. 1; Supplementary Data 1, see also "Methods"). PBMCs were stimulated with LPS or TNF and phopho-p65 was measured as $log_{10}$ fold change (FC) median fluorescence intensity (MFI) in CD4+ T cells, CD8+ T cells, NK, NKT cells, CD14+ monocytes, CD16+ monocytes, intermediate monocytes, plasmacytoid dendritic cells, CD141+ type 1 conventional DC (cDC1), CD1c+ type 2 conventional DC (cDC2) and double negative DC (dnDC) (Fig. 2a). In keeping with the assay being performed in isolated PBMCS rather than in whole blood, and thus in absence of free adalimumab in the blood, we did not detect any inhibitory effect of adalimumab on TNF-induced NF-κB phosphorylation during therapy, in neither lymphoid or myeloid cells in the PSORT adalimumab combined cohort ($n = 43$) (Supplementary Fig. 5 and Supplementary Data 5). Next, we evaluated NF-κB-p65 phosphorylation in DC subsets within PBMCs of the PSORT adalimumab discovery cohort ($n = 16$) previously assayed for NF-κB nuclear translocation. In keeping with the correlation

between residual disease and NF-κB nuclear translocation in DCs (Fig. 1d), we detected a statistically significant correlation ($r^2 = 0.714$, $p = 3 \times 10^{-4}$, FDR < 0.01, linear regression $t$ test) between LPS-induced NF-κB phosphorylation in cDC2 at baseline and residual disease at w12 (Fig. 2b and Supplementary Data 6), thus validating our previous finding using a different, but biologically related, analytical read-out. Moreover, LPS-induced NF-κB phosphorylation at w0 was significantly lower in cDC2 of PASI75 responding patients ($p < 0.05$, Mann–Whitney $U$ test; Fig. 2c), with a similar downward trend also observed in patients reaching PASI90 (Supplementary Fig. 6a). NF-κB phosphorylation in cDC2 significantly correlated with NF-κB nuclear translocation observed in total DC of the same patients ($r^2 = 0.586$, $p = 0.099$, linear regression $t$ test; Fig. 2d), further validating cDC2 as the main DC subset implicated in clinical response to adalimumab.

Next, we sought to replicate our findings in DCs in the PSORT adalimumab replication cohort, ($n = 27$; Supplementary Fig. 1). LPS or TNF-induced NF-κB phosphorylation in DC subsets was correlated to clinical response at w12. We observed nominal correlations (all linear regression $t$ test, FDR > 0.05) between residual disease and NF-κB phosphorylation induced by either LPS ($r^2 = 0.347$, $p = 0.0209$) or TNF ($r^2 = 0.425$, $p = 0.0062$) in cDC2, as well as TNF-induced NF-κb phosphorylation in pDC ($r^2 = 0.502$, $p = 0.0021$) at w0 (Supplementary Fig. 6b and Supplementary Data 6). Nevertheless, NF-κB phosphorylation induced by LPS and TNF in cDC2, and by TNF in pDC at w0 was significantly lower in PASI75 responders than in PASI75 non-responders ($p < 0.01$, Mann–Whitney $U$ test; Supplementary Fig. 6c). Finally, we detected statistically significant correlations between NF-κB phosphorylation induced by either TNF ($r^2 = 0.285$, FDR = 0.029), or LPS ($r^2 = 0.573$, FDR = $10^{-4}$) in cDC2, as well by TNF in pDCs ($r^2 = 0.245$, FDR = 0.043) and residual disease at w0 (all linear regression $t$ test, Fig. 2e and Supplementary Data 6) in the PSORT adalimumab combined cohort ($n = 43$, Fig. S1). These correlations were not driven by any clinical covariate previously associated with response to adalimumab such as age, gender, ethnicity, smoking, weight, psoriatic arthritis, being biologic naive, the baseline PASI or the presence of the HLA-C*06:02 allele[8,10] (Supplementary Data 7). In keeping with the correlation analysis, NF-κB phosphorylation at baseline was significantly lower in LPS ($p < 0.0001$, Mann–Whitney $U$ test) and TNF ($p < 0.01$, unpaired $t$ test)-stimulated cDC2 and TNF-stimulated pDC ($p < 0.0001$, Mann–Whitney $U$ test) in PASI75 responders than non-responders (Fig. 2f–g), with a similar trend for most stimulations for PASI90 responders and non-responders (Supplementary Fig. 6d). Interestingly, NF-κB phosphorylation induced in PASI75 responders was the same level as HV, while it was higher in non-responders (Supplementary Fig. 6e). The correlation between NF-κB phosphorylation and residual disease was progressively lost in the presence of adalimumab at w 1 and w12 (Supplementary Data 6), albeit TNF-induced ($p < 0.05$, Mann–Whitney $U$ test) and LPS-induced ($p < 0.05$, Mann–Whitney $U$ test) NF-κB phosphorylation in cDC2 was still significantly higher in PASI75 non-responders at w1 (Supplementary Fig. 6f).

Thus, regardless of the stimulus, NF-κB activation in cDC2 before therapy plays a role in determining clinical response to adalimumab and may have a role as a predictive biomarker of response.

**moDCs from PASI75 non-responders have increased maturation at baseline.** Next, we sought to gain some mechanistic insights into the correlation between NF-κB activation in cDC2 cells and the lack of clinical response to adalimumab. It is well established that NF-κB activation drives DC maturation through

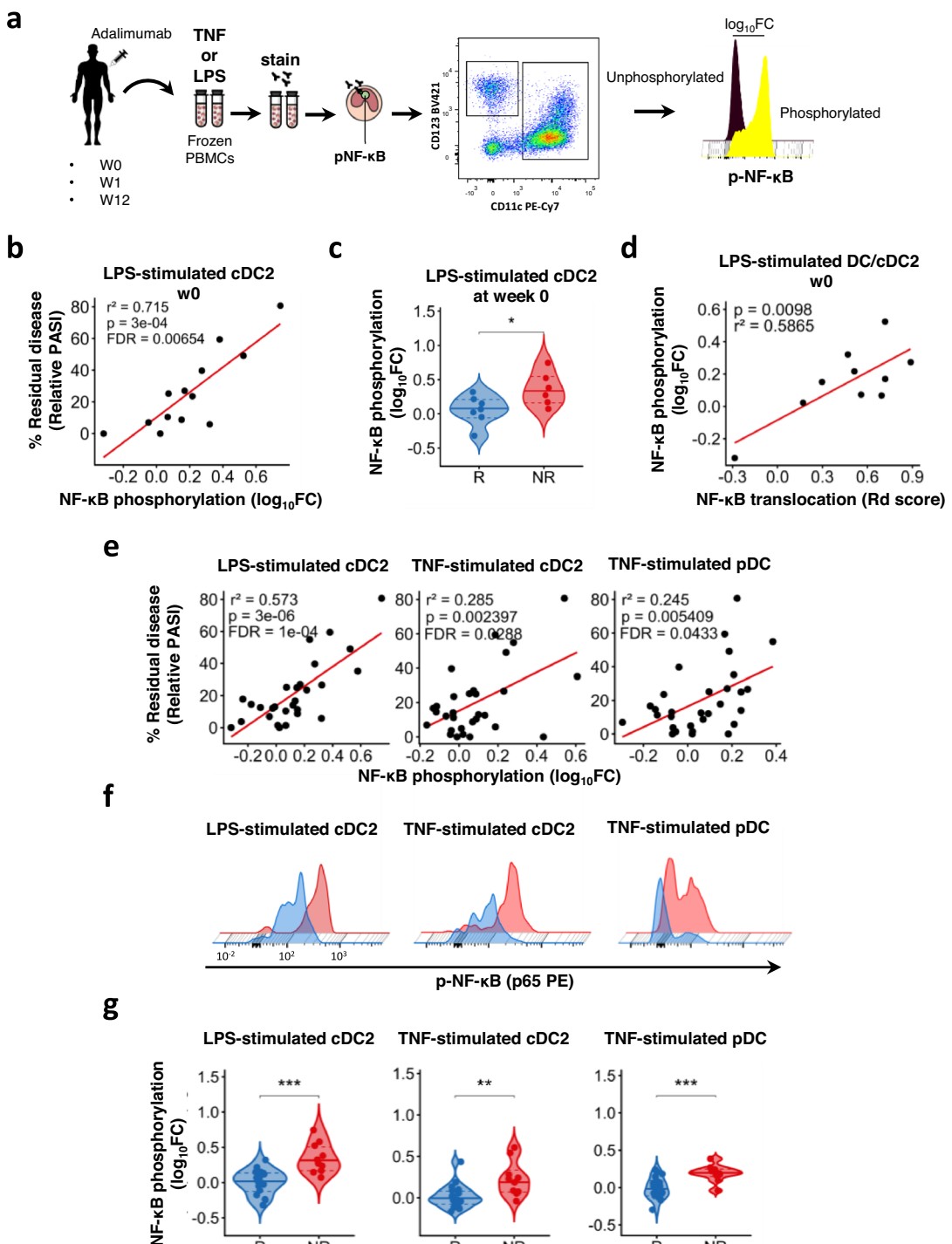

**Fig. 2 LPS-induced NF-κB phosphorylation in cDC2 at baseline correlates with response to adalimumab. a** PBMC from psoriasis patients obtained at baseline (week 0, w0) and at w1 and w12 after starting adalimumab therapy were stimulated with either TNF or LPS and NF-κB p65 phosphorylation was measured by phospho flow cytometry. **b** Correlation analysis between LPS-induced NF-κB p65 phosphorylation ($\log_{10}FC$) in conventional Type 2 DC (cDC2) at w0 and clinical response, expressed as a residual disease at w12 and measured as relative PASI in the discovery cohort ($n = 13$). Each dot represents one patient. **c** Violin plot graphs of LPS-induced NF-κB p65 phosphorylation in cDC2 at w0 in PASI75 adalimumab responders PASI (R, blue, $n = 7$) and non-responders (NR, red, $n = 6$) *$p = 0.035$. **d** Correlation between LPS-induced NF-κB nuclear translocation in DC and LPS-induced NF-κB p65 phosphorylation in cDC2 at w0. **e** Correlation analyses between NF-κB p65 phosphorylation induced by different stimuli in cDC2 and plasmacytoid Dc (pDC) and residual disease in the PSORT adalimumab combined cohort ($n = 25$–30). **f** Representative phospho flow overlay histograms of NF-κB p65 phosphorylation in DC subsets in PASI75 adalimumab responders (blue) and non-responders (red). **g** Violin plot graphs of NF-κB p65 phosphorylation induced by various stimuli in DC subsets in PASI-75 adalimumab responders (blue, $n = 16$–20) and non-responders (red, $n = 9$–10) ($p$ cDC2-LPs $= 0.00056$, $p$ cDC2-TNF $= 0.013$, $p$ pDCs-TNF $= 0.00034$). $p$ values and FDR are reported on the graph in (**b**), (**d**) and (**e**); Mann–Whitney $U$ test (**c**, **g**, left and right panels) or unpaired $t$ test (**g** middle panel). All tests are two-sided. Source data are provided as a Source Data file.

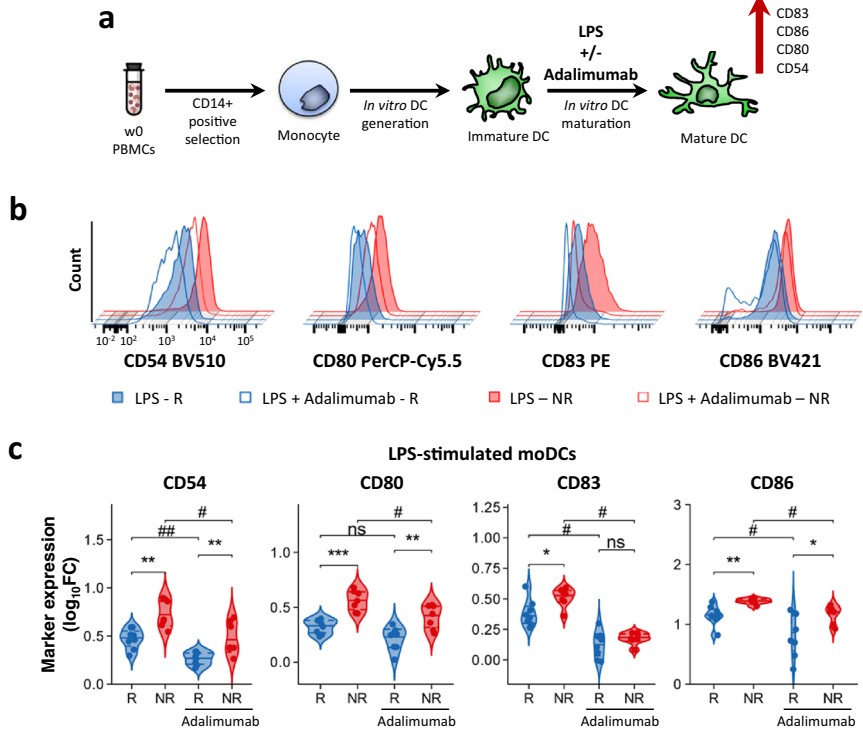

**Fig. 3 LPS induces increased moDC maturation in adalimumab non-responders. a** moDCs were generated in vitro from monocytes obtained from w0 PBMCs. Maturation was induced by adding LPS in the presence or absence of adalimumab. **b** Representative flow overlay histograms of maturation markers expression in LPS-matured (filled) and LPS + adalimumab (empty) moDCs generated in PASI75 adalimumab responders (R, blue) and non-responders (NR, red). **c** Violin plot graphs of maturation markers expression (measured as log10FC MFI vs Immature DCs) in PASI-75 adalimumab responders (blue, $n = 8$) and non-responders (red, $n = 7$). ($p$ CD54 RvsNR = 0.0012, RvsNR A = 0.0037; # CD54 = 0.0156, ## CD54 = 0.0078; $p$ CD80 RvsNR = 0.0003, RvsNR A = 0.0037; # CD80 = 0.0156; $p$ CD83 RvsNR = 0.04; # CD83 RvsR-A = 0.0156, NRvsNR-A = 0.0156; $p$ CD86 RvsNR = 0.0022, RvsNR A = 0.02; # CD86 RvsR-A = 0.0391, NRvsNR-A = 0.0156), Mann–Whitney U test (**c**) between R and NR. ## $p < 0.01$, # $p < 0.05$, Two-sided Wilcoxon matched-pair rank test for adalimumab effect. Source data are provided as a Source Data file.

the expression of co-stimulatory molecules critical for T cell priming and activation[18]. Thus, we hypothesized that the increased NF-κB activation induced by inflammatory stimuli in cDC2 in patients failing adalimumab therapy may reflect an increased propensity to upregulate the expression of co-stimulatory molecules.

To test our hypothesis, we generated monocyte-derived immature DCs in vitro, as model for cDC2, using cryopreserved baseline PBMCs of PASI75 responding and non-responding patients ($n = 17$) and induced DC maturation with LPS, in the presence or absence of adalimumab (Fig. 3a). Mature moDC of PASI75 non-responders displayed a significantly increased expression of CD54 ($p = 0.0012$, all p values are from a Mann–Whitney U test), CD80 ($p = 3.11 \times 10^{-4}$), CD83 ($p = 0.04$) and CD86 ($p = 0.0087$), and an upward trend for CD40 and HLA-DR as compared to responders, with similar expression in responders and HV and increased level in non-responding patients (Fig. 3b–c and Supplementary Fig. 7a). While the addition of adalimumab significantly inhibited LPS-induced DC maturation in the overall cohort (Supplementary Fig. 7b), as well as in responders and non-responders (Fig. 3c), the levels of CD54 ($p = 0.009$, all p values are from a Wilcoxon test), CD80 ($p = 0.003$) and CD86 ($p = 0.02$) remained significantly higher in PASI75 non-responders as compared to responders also in presence of adalimumab (Fig. 3b–c). moDC maturation in the presence of adalimumab significantly reduced cell survival, however, no difference was observed in PASI75 responding and non-responding patients (Supplementary Fig. 7c, d). Thus, moDC of PASI75 non-responders display an increased propensity to

upregulate co-stimulatory molecules in vitro which is not offset by the inhibitory effect exerted by adalimumab, in line with the effect exerted by adalimumab therapy on NF-kB translocation in blood DC (Fig. 1g). The increased moDC maturation was accompanied by a significant increase and an upward trend in the production of IL12 and IL23 respectively, in the culture supernatants of mature moDC of PASI75-non-responders, as compared to PASI75-responders, whereas there was no difference in the secretion of IL10, TNF, IL1β and IL6 (Supplementary Fig. 8). Taken together, lack of clinical response to adalimumab is associated with the intrinsic increased maturation and activation potential of DC both in the presence and absence of the drug.

**Frequency of IL-17⁺ T cells in the blood is associated with response to adalimumab.** DC maturation leads to T cell activation and cytokine production. Thus, we investigated whether the increased propensity of adalimumab non-responders DCs to upregulate co-stimulatory molecules in vitro and their resistance to the inhibitory effect of adalimumab (Figs. 3, 1g) has an effect on T cell responses and can be detected in circulating DCs. To this end, we studied the phenotype of T cells and DCs in the blood of psoriasis patients before and during adalimumab therapy.

First, we studied circulating T cells producing the psoriasis hallmark cytokine IL-17A alongside IL-10 producing T regulatory (Treg) in psoriasis patients and HV (Supplementary Fig. 9). As previously reported[19], frequency of IL17A⁺ CD4⁺ T (Th17), IL17A⁺ CD8⁺ T (Tc17) and IL-17A⁺ Treg cells was significantly increased at baseline in PBMCs of psoriasis patients as compared

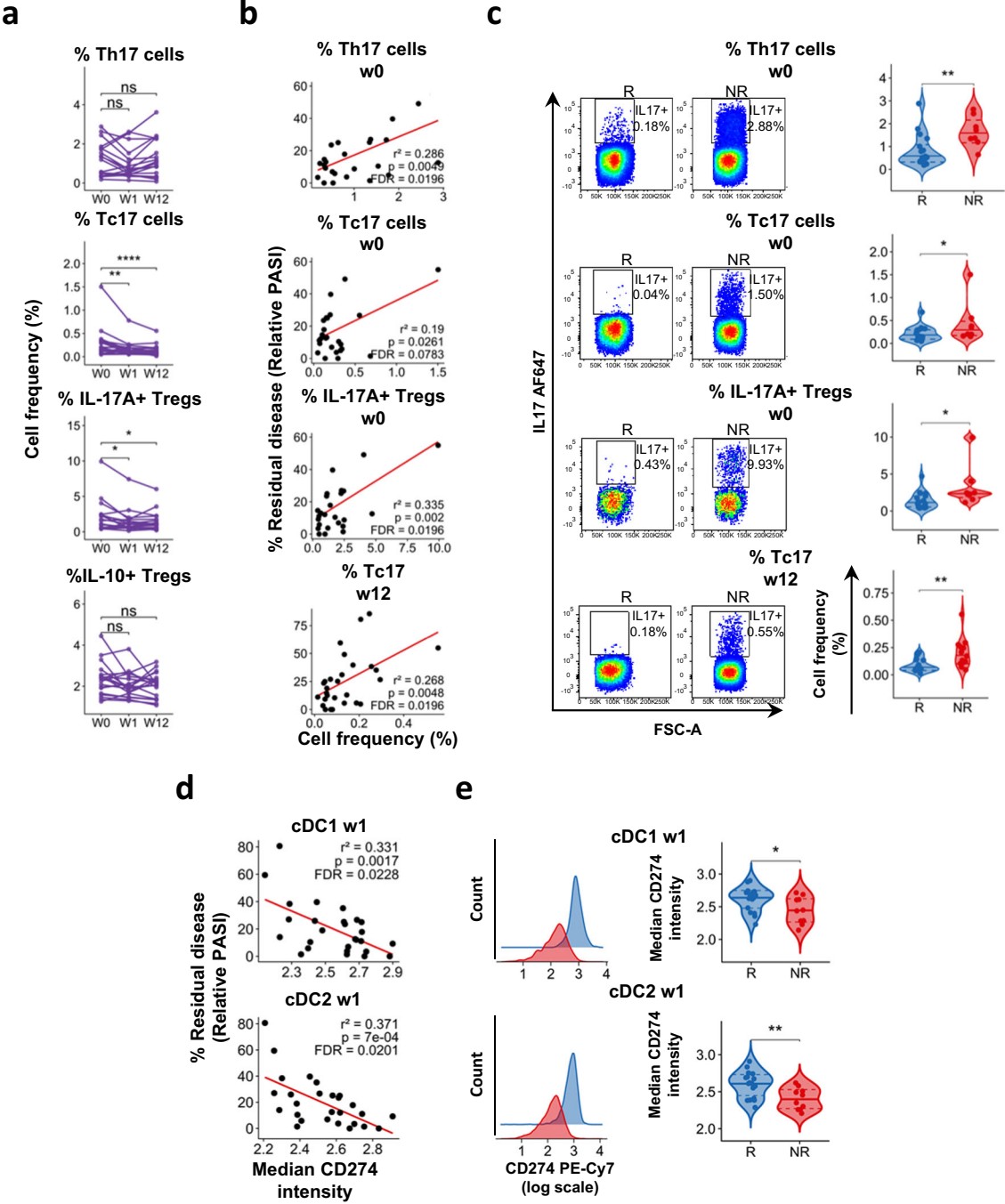

**Fig. 4 Response to adalimumab correlates with the frequency of IL17A+ T cells and CD274 expression in DC.** Immunophenotyping of PBMCs of psoriasis patients at w0 and at w1 and w12 after starting adalimumab therapy. **a** Frequency of IL10+ and IL17A+ CD4, CD8 T cells and Tregs at different time points, each line represents one patient ($n = 19$) (p Tc17 W0vsW1 = 0.0014, W0vsW12 = $3.8 \times 10^{-6}$; IL17A Tregs W0vsW1 = 0.045, W0vsW12 = 0.018). **b** Correlation analyses between the frequency of IL17A+ or IL10+ cells within CD4, CD8 or Tregs and residual disease at w12 ($n = 26$–28). **c** Representative flow cytometry colour-plots (left) and violin plots graphs of the frequency of cytokine-producing cells in adalimumab PASI75 responders (R, blue, $n$ week 0 = 18, $n$ week 12 = 15) and non-responders (NR, red, $n$ week 0 = 8, $n$ week 12 = 13) (p Th17 w0 = 0.0051, Tc17 w0 = 0.047, IL17A Tregs w0 = 0.011, Tc17 w12 = 0.0037). **d** Correlation analyses between the expression of CD274 at w1 and residual disease at w12 in cDC1 and cDC2 ($n = 27$). **e** Representative histograms overlay graphs (left) and violin plots graphs (right) of CD274 expression in DC of PASI75 adalimumab responders (blue, $n = 17$) and non-responders (red, $n = 10$) *$p = 0.045$, **$p = 0.0058$. Kruskal–Wallis with Dunn's multiple comparisons post-test (**a**), Mann–Whitney $U$ test (**c**) or unpaired $t$ test (**e**). All tests are two-sided. Source data are provided as a Source Data file.

to healthy volunteers (Supplementary Fig. 10). Tc17 and IL-17A+ Treg cells significantly decreased during adalimumab therapy, while frequency of Th17 and IL10+ Treg cells did not change (Fig. 4a). Moreover, we detected a statistically significant correlation (liner regression) between residual disease and the

frequency of Th17 and IL-17A+ Treg at w0 and of Tc17 at w12 (Fig. 4b and Supplementary Data 8). In keeping with this finding, PASI75 non-responders had significantly higher frequency of Th17 and IL-17A+ Treg at w0 and Tc17 at week 12 (Fig. 4c). Moreover, the frequency of Th17 and IL-17A+ Tregs cells was

similar in HV and PASI75 responders, but significantly higher in non-responders (Supplementary Fig. 10b). On the other hand, the frequency of Tc17 cells at w0 was significantly higher in both responders and NR, but decreased to HV level in PASI75 responders at w12. Thus, clinical response to adalimumab is associated with a decrease in circulating Tc17 at week 12, in keeping with the emerging critical role for CD8+ T cells in psoriasis[20,21]. Moreover, the frequency of Th17 cells prior to therapy may have a role as a predictive biomarker of response, in line with the heightened NF-kB activation and maturation status induced in DCs.

**Response to adalimumab is associated with early upregulation of CD274 in DCs.** Next, we deeply phenotyped blood DC subsets (Combined PSORT adalimumab cohort, $n = 43$) using a 15-colours flow-cytometry panel comprehensive of activation (HLA-DR, CCR7, CD40), maturation (CD80, CD83, CD86), adhesion (CD54, CD209) and inhibitory (CD274) markers, and applied Flow self-organizing map (SOM) unsupervised clustering analysis[22,23] to identify cell populations and DC subsets (Supplementary Fig. 11a). The frequency of cDC1 was significantly decreased at week 12 of adalimumab therapy, while cDC2 and pDC were unaffected (Supplementary Fig. 11b) but neither of them correlated at any time point with clinical response at week 12 (Supplementary Fig. 11c). Overall, marker expression did not change over time (Supplementary Fig. 12 and Supplementary Data 9), or correlate, at any time point, with clinical response at week 12 (Supplementary Data 10) or differ between PASI75 responders and non-responders (Supplementary Data 11). Nevertheless, we detected statistically significant correlations (linear regression) between relative PASI and CD274 expression at week 1 in cDC1 ($r^2 = 0.331$, FDR < 0.05) and cDC2 ($r^2 = 0.371$, FDR < 0.05) (Fig. 4d and Supplementary Data 10) and CD274 expression in cDC1 ($p < 0.05$, unpaired t test) and cDC2 ($p < 0.01$, unpaired $t$ test) at week 1 was nominally different (FDR > 0.05) in PASI75 responders and non-responders (Fig. 4e and Supplementary Data 11).

To explore a possible genetic influence on CD274 expression, we mined the Genotype-Tissue Expression database (GTEX, www.gtexportal.org) and identified rs59906468 as a strong cis eQTL (normalized effect size = 0.26, $p = 3.6 \times 10^{-25}$, linear regression $t$ test) in whole blood, with the A allele reducing the expression of *CD274* (Supplementary Fig. 13a). We observed a similar trend for CD274 protein expression in DC subsets at various time points in our genotyped adalimumab cohort (Supplementary Fig. 13b). Moreover, in an extended PSORT genetic dataset[10], several putative associations with adalimumab response were observed in the *CD274* region at both 3 and 6 m for both PASI75 and PASI90 response outcomes (*p*-values from 0.10 to $8.8 \times 10^{-6}$ logistic regression $t$ test Supplementary Fig. 13c). Bayesian colocalization analysis suggested that rs59906468 may be a shared causal variant for both CD274 protein expression and adalimumab response, particularly PASI90 response at 6 m (posterior probability for shared effect = 0.66; Supplementary Fig. 13d).

Thus, deep phenotyping of blood DCs suggests that clinical response to adalimumab is associated with early upregulation of the inhibitory molecule CD274 in DCs, which may be genetically driven. However, the surface phenotype of blood DCs prior to therapy does not play a major role in determining clinical response nor has biomarker potential.

**Increased DC maturation and IL-17+ T cell activation in psoriasis skin of non-responders at baseline.** DC maturation is triggered at sites of inflammation and mature DCs are abundant in psoriasis lesions. Thus, the heightened NF-kB activation and maturation propensity detected in blood DCs following ex vivo and in vitro stimulation may have an effect on the surface phenotype of skin rather than circulating DCs. To this end, we studied the effect of adalimumab on DC, as well as, T cell phenotype in lesional skin sections of PASI75 responders and non-responders at w0 and w12 ($n = 20$). Baseline disease severity did not differ in responders (median PASI at w0 = $12.9 \pm 3.1$) and non-responders (median PASI at w0 = $13.0 \pm 4.7$). Adalimumab therapy significantly decreased the number of CD11c+ DC ($p < 0.01$, paired $t$ test) and their expression of CD83 ($p < 0.01$, paired $t$ test), as well as the number of IL23+ ($p < 0.01$ Wilcoxon test) DC in all patients at week 12 (Fig. 5a and Supplementary Fig. 14). However, PASI75-non-responders harboured an increased number ($p < 0.05$, Mann–Whitney $U$ test) of CD11c+ DC which had enhanced expression of CD83 ($p < 0.05$, unpaired $t$ test) and produced IL23 in greater number ($p < 0.05$, unpaired $t$ test) as compared to PASI75-responders at w0 (Fig. 5b, c), confirming a more mature skin DC phenotype in non-responders, in line with the DC maturation data obtained in vitro (Fig. 3). While PDL-1 expression in CD11c DC increased with therapy in the overall cohort (Supplementary Fig. 14b), no difference was observed between responders and non-responders at any time point (Supplementary Fig. 14c, d). As expected, the number of IL17+CD4+ (Th17, $p < 0.01$, paired $t$ test) and IL17+CD8+ T (Tc17, $p < 0.05$, paired t test) cells present in psoriatic skin at w12 (Fig. 5d) was decreased by adalimumab therapy. The number of Th17 ($p < 0.05$, Mann–Whitney $U$ test) and Tc17 ($p < 0.05$, Mann–Whitney $U$ test) cells was significantly higher in the skin of non-responders at w0 (Fig. 5e, f). Of interest, while number of Th17 at w12 was similar in responders and non-responders, Tc17 cells remained significantly elevated ($p < 0.01$, Mann–Whitney $U$ test) in the skin of non-responders as compared to responders (Fig. 5e, f), in line with the association between clinical response to adalimumab and the decrease in circulating Tc17 cells observed in blood at w12 (Fig. 4).

**NF-κB phosphorylation in cDC2 at baseline is a biomarker of response to adalimumab.** Finally, we evaluated the predictive value for patient stratification of the 8 blood immune traits measured at w0 or w1 which significantly correlated with clinical response at w12 (i.e. NF-κB phosphorylation at w0 induced by LPS in cDC2 and by TNF in pDC and cDC2; frequency of Th17, Tc17 and IL-17+ Tregs at w0, CD274 expression at w1 in cDC1 and cDC2). We built receiver operating characteristic (ROC) curves for the binary outcome of PASI75 responders or non-responders at w12 for each potential biological read-out (NF-κB phosphorylation, T cell frequency, CD274 expression) obtained in the PSORT adalimumab combined cohort (Supplementary Fig. 15a) and ranked them according to their area under the curve (AUC). The top markers were: LPS-induced NF-κB phosphorylation in cDC2 at w0, frequency of Th17 at w0 and expression of CD274 in cDC2 at w1, with an AUC of 0.922, 0.840 and 0.806, respectively (Supplementary Fig. 15a and Fig. 6a). The best performing biomarker overall, NF-κB phosphorylation in cDC2 (cut-off threshold of 0.169 $\log_{10}$FC) had 80% sensitivity, 88.9% specificity, 85.7% accuracy, and predictive odds ratio (Eq. (2)) of 32 (Fig. 6b). AUC did not improve when we combined LPS-induced NF-κB phosphorylation in cDC2 with the other top signals identified (Fig. 6c), therefore we selected this biomarker for further investigation and validation. The psoriasis-susceptibility HLA-C*06:02 allele has been associated with clinical response to biologics, with carriers less likely to respond to adalimumab[10]. However, the inclusion of HLA-C*06:02 genotype in our models did not increase the predictive value of LPS-induced NF-κB phosphorylation in cDC2 (Fig. 6d). Follow-up data for 26 of the

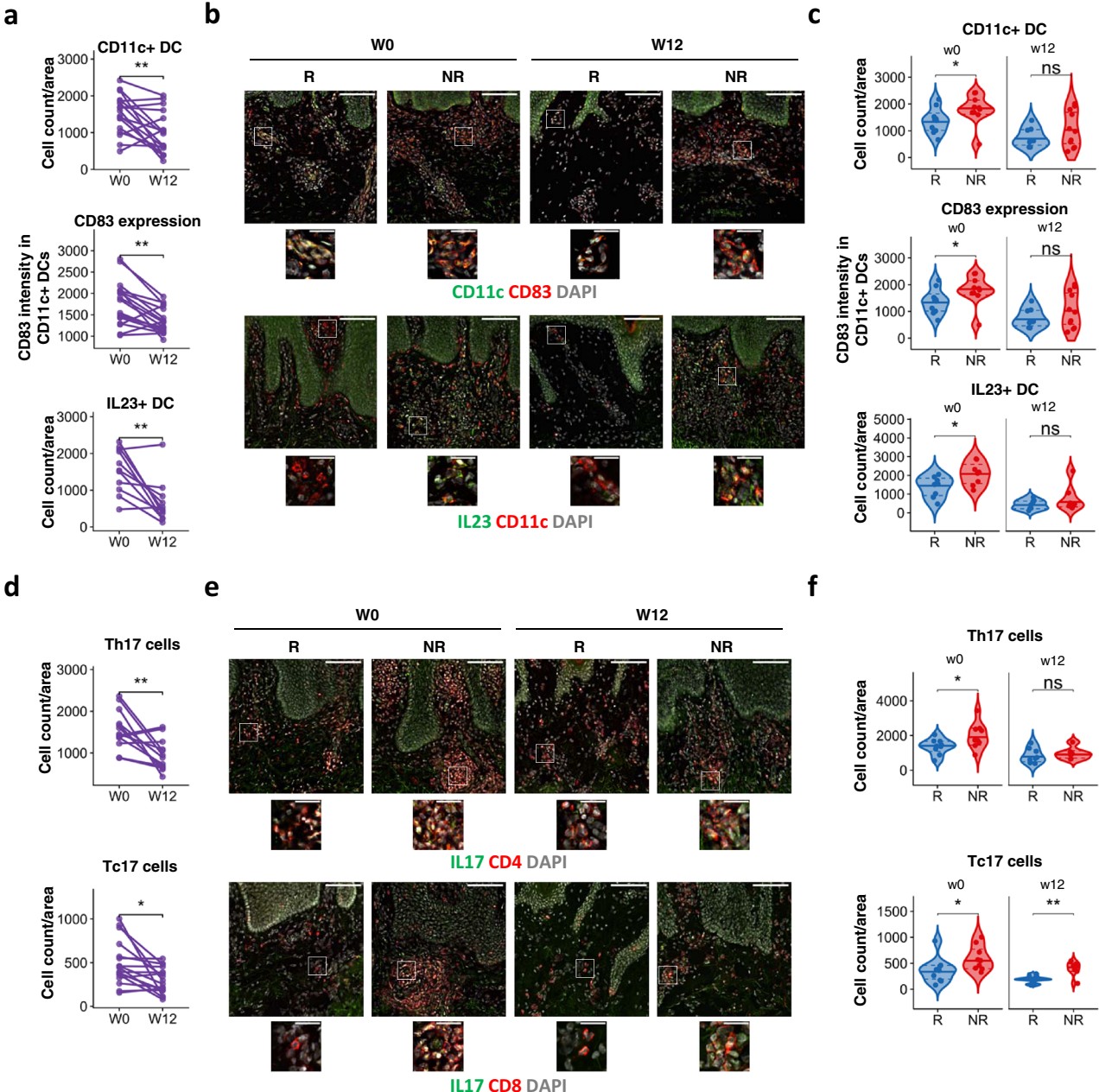

**Fig. 5 Response to adalimumab is associated with DC and IL-17$^+$ T cell activation.** Analysis of psoriasis lesional skin of patients at baseline (w0) and after 12 weeks (w12) of adalimumab treatment a Number of CD11c+ DCs, expression of CD83 in CD11c+ dermal dendritic cells (DC) and number of IL-23+ CD11c+ DC per area of analysis (750 × 750 μm) at different time points. Each line represents one patient (n CD11c and CD83 = 17, n IL23 = 11) (p CD11c = 0.0093, CD83 = 0.0012, IL23 = 0.0049). **b** Representative immunofluorescence images of psoriasis skin showing total number of CD11c+ DCs (top), CD83 expression in CD11c+ DC (middle) and IL23+ CD11c+ DC (bottom) in PASI75 adalimumab responders (R, n = 8–10) and non-responders (NR, n = 6–10). **c** Violin plot graphs of CD11c+ DC counts (top), CD83 expression in dermal DC (middle) and number of IL23 + DC in adalimumab responders (blue, n = 8–10) and non-responders (red, n = 6–10) at w0 and w12) (p CD11c = 0.043, CD83 = 0.035, IL23 = 0.039). d Number of IL-17A +CD4+ Th17 and IL-17A+CD8+ Tc17 cells per area of analysis at different time points. Each line represents one patient (n Th17 = 13, n Tc17 = 17) (p Th17 = 0.0013, Tc17 = 0.011). **e** Representative immunofluorescence images of psoriasis skin showing Th17 (top) and Tc17 (bottom) in adalimumab responders (n = 8–10) and NR (n = 6–9). **f** Violin plot graphs of the number of Th17 and Tc17 cells in lesional skin sections of in adalimumab responders (blue, n Th17 week 0 = 9, n Th17 week 12 = 8, n Tc17 week 0 = 10, n Tc17 week 12 = 9) and non-responders (red, n Th17 week 0 = 9, n Th17 week 12 = 6, n Tc17 week 0 = 9, n Tc17 week 12 = 8) at w0 and w12 (p Th17 = 0.04, Tc17 w0 = 0.028, Tc17 w12 = 0.0037). Wilcoxon test (**a** bottom panel), or paired t test (**a** except bottom panel, **d**), Mann–Whitney U test (**c** top panel, **f**) or unpaired t test (**c** except top panel). All tests are two-sided. Scale bars in (**b**, **e**): 100 μm (overview) and 25 μm (insets). Source data are provided as a Source Data file.

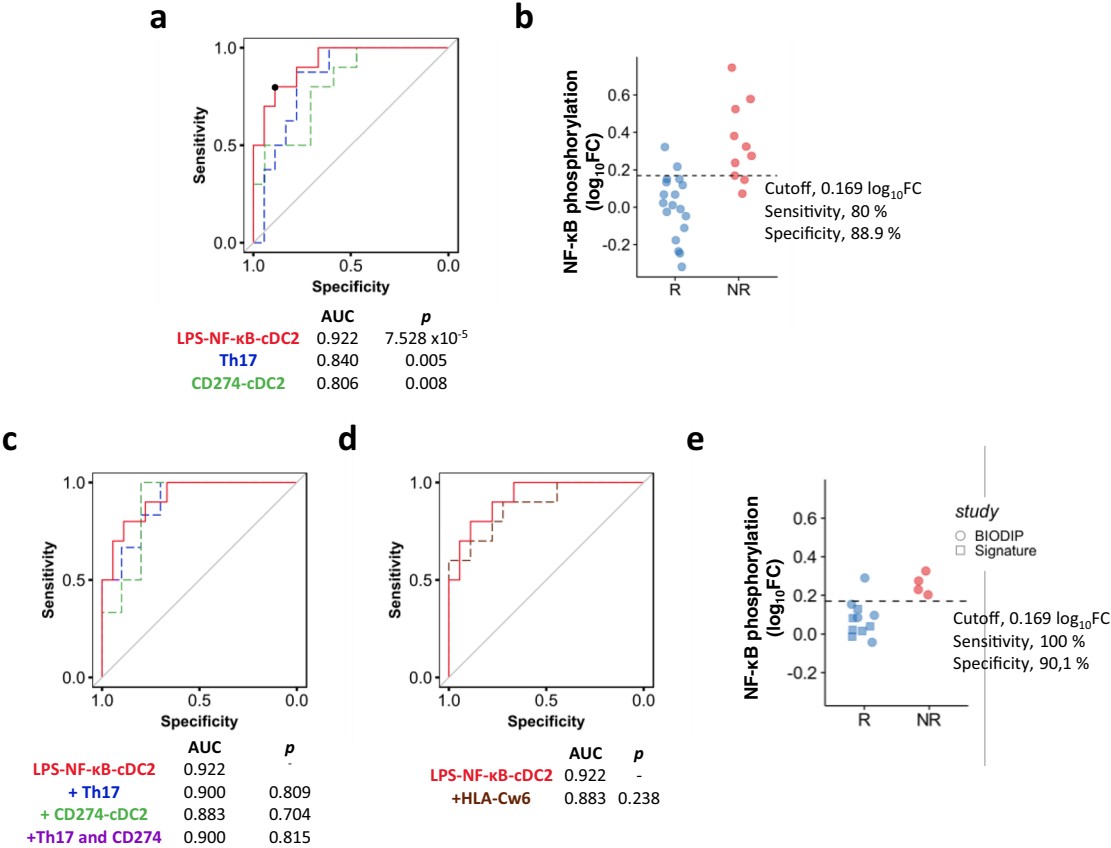

**Fig. 6 NF-κB phosphorylation in cDC2 at baseline as biomarker of response to adalimumab. a** Receiver operator characteristic (ROC) of linear models for LPS-induced NF-κB phosphorylation in cDC2 at baseline (w0, red line), frequency of blood Th17 cells at baseline (w0, blue line) and PDL1 expression in cDC2 at w1 (green line) predicting PASI75 outcome in adalimumab patients at week 12 (combined cohort, $n = 26$–28). Black dot shows the cut-off for maximal accuracy for the best ROC curve. $p$ values vs an AUC of 0.5 are shown on the graph. **b** Dot plot graph of the PSORT adalimumab combined cohort ($n = 28$) classified in PASI75 responder (R, blue, $n = 18$) and non-responders (NR, red, $n = 10$) according to the cut-off for LPS-induced NF-κB phosphorylation in cDC2 at w0. **c** ROC of LPS-induced NF-κB phosphorylation in cDC2 at w0 (solid red line) and combination biomarkers (LPS-NF-kB-cDC2 + Th17, dotted blue line; LPS-NF-kB-cDC2 + CD274-cDC2 dotted green line, and LPS-NF-kB-cDC2 + Th17 + CD274-DC2, dotted purple line). The purple line partially overlaps with the blue line. $p$ values vs AUC of LPS-NF-kB-cDC2 are shown. d ROC analysis for LPS-NF-kB-cDC2 predicting PASI75 outcome in adalimumab patients (solid red line) and when combined with HLA-C*06:02 genotype (dotted brown). $p$ value vs AUC of LPS-NF-kB-cDC2 is shown. e ROC clinical validation using optimal cut-off for LPS-induced NF-κB phosphorylation in cDC2 at w0 identified in the PSORT combined cohort to classify an independent cohort of adalimumab patients in PASI75 responders (blue, $n = 11$) and non-responders (red, $n = 4$) (clinical validation cohort). Two-tailed Mann–Whitney U test, (**a**, **c**, **d**). Source data are provided as a Source Data file.

28 patients assessed at w12, showed that LPS-induced NF-κB phosphorylation in cDC2 at w0 also predicted PASI75 response at 6 months (cut-off threshold 0.151 $log_{10}FC$, 85.7% sensitivity, 73.7% specificity and AUC of 0.80 ($p < 0.05$, Mann–Whitney U test) (Supplementary Fig. 15b).

Finally, we tested the ability of the model to predict clinical response to adalimumab at w12 in a fully independent cohort of 15 psoriasis patients (clinical validation cohort, Supplementary Fig. 1). Applying the previously determined (Fig. 6b) cut-off threshold of 0.169 $log_{10}$ FC for LPS-induced NF-κB phosphorylation in cDC2, the model predicted treatment outcome to adalimumab at w12 with 93.3% accuracy, 100% sensitivity 90.1% specificity (Fig. 6e).

Taken together, our results clearly identify TNF-induced and LPS-induced NF-κB phosphorylation in cDC2 dendritic cells prior to therapy as a predictive biomarker of response to adalimumab which may aid in patient stratification.

## Discussion

Here we report the discovery and replication of an actionable, predictive biomarker of response to the anti-TNF adalimumab in

psoriasis. NF-κB activation of blood cDC2, measured as LPS-induced NF-κBp65 phosphorylation before therapy, correlated with clinical outcome independently of covariates such as disease severity, and discriminated between PASI responders and non-responders with high sensitivity and specificity for up to 6 months after commencing therapy. Moreover, we provide some insights into the downstream cellular and molecular events associated with clinical response to adalimumab. The increased induced NF-kB activation detected in non-responders before commencing adalimumab was associated with increased DC maturation and resistance to the inhibitory effect of adalimumab in vitro, as well as well increased IL-17+ T cell activation in blood, and ultimately a more mature and proinflammatory DC phenotype and increased number of IL-17+ T cells in the skin.

NF-κB signalling plays a critical role in inflammation and multiple genes of this pathway are associated with psoriasis susceptibility[24]. Activation of NF-κB by inflammatory stimuli, through the engagement of TLRs and cytokine receptors, culminates in the translocation of NF-κB complexes into the nucleus and activation of gene expression. Phosphorylation of p65 NF-κB subunit, such as at Ser529, further regulates NF-κB activation and

function by increasing its transcriptional activity[17]. We show that ex vivo TNF-induced NF-κB translocation was largely inhibited in most immune cell types by the free drug present in the blood of patients sampled during therapy. Nevertheless, the near-complete inhibition of induced NF-κB signalling in T cells mediated by adalimumab does not underpin its mechanism of action in psoriasis as it does not correlate with clinical response. On the other hand, TNF-induced NF-kB activation in DC was only partially inhibited by adalimumab, with DC from non-responders more refractory to its inhibitory effect. Moreover, NF-κB activation induced in DC by various pro-inflammatory stimuli before commencing therapy, and measured as either nuclear translocation or p65 phosphorylation, correlated with lack of clinical response at w12.

DC of myeloid and plasmacytoid lineage are highly implicated in psoriasis pathogenesis, alongside T cells and keratinocytes[2,3]. Mature myeloid cDC with high T-cell stimulatory capacity are enriched in skin lesions where they produce IL-23 and other proinflammatory cytokines[25] and are poised to present putative auto-antigens, such as LL37, to T cells. Activation of NF-κB signalling in DC, triggered by either pathogen-associated molecular patterns or proinflammatory cytokines, induces most of the phenotypic and functional characteristics typical of mature DCs, inducing the expression of MHC II and co-stimulatory molecules[26,27]. TNF blockade by anti-TNF etanercept and infliximab impairs maturation of DC generated in vitro from healthy volunteers, resulting in reduced levels of HLA-DR and co-stimulatory molecules[16,28]. Moreover, the expression of co-stimulatory molecules in DCs from psoriasis skin diminished during etanercept therapy[16] and in vitro-derived DC from rheumatoid arthritis patients receiving anti-TNF displayed impaired upregulation of co-stimulatory molecules and poor T-cell stimulatory activity[28]. Our data build on these previous findings, validating cDC2 as the key mechanistic cellular target of adalimumab influencing the clinical outcome. cDC2 are the major population of myeloid cDC in human blood, produce high levels of IL-23 and are potent activator of Th17 and CD8+ T cells[29]. Moreover, cDC2, but not cDC1, are largely increased in psoriasis lesional epidermis as compared to non lesional and healthy skin[30]. Thus, the biomarker potential of this specific subset is in line with their major involvement in psoriasis.

In keeping with the increased NF-κB activation, the phenotype of in vitro-generated DC in PASI75 non-responders before commencing adalimumab was strikingly different from DC generated from responders and healthy volunteers. Non-responders DCs displayed a more mature phenotype, which albeit inhibited by in vitro adalimumab, remained significantly different from that of responders, suggesting that an intrinsic increased propensity to respond to inflammatory and maturation stimuli limits the beneficial effect of TNF-blockade. Crucially, and despite disease severity was comparable between the two groups, the skin of PASI75 non-responders harboured DC with a more mature and activated phenotype before commencing the therapy, thus validating the in vitro findings. Moreover, the lack of phenotypic differences in circulating blood DC prior to therapy suggests that a pro-inflammatory environment is required to trigger such differences, otherwise undiscernible in resting ex vivo cells.

The significant correlation between clinical response and NF-κBp65 phosphorylation in cDC2, induced by TNF or LPS suggests a mechanistic link between DC intrinsic maturation potential and adalimumab clinical efficacy. Not surprisingly, the correlation with the best predictive value was detected in LPS-stimulated cells, as LPS is more efficient than TNF to activate NF-kB in cDC2 in our experiments and has been shown to be more efficient at maturing DCs in vitro[31]. Nevertheless, LPS induces

autocrine TNF in DC[32], thus providing a further potential biological rationale between the identified biomarker and the biological pathway targeted by adalimumab.

DC immunogenicity is determined both by their maturation state and their lifespan. moDCs and blood cDC2 express both TNF receptor (TNFR)-1 and TNFR2, withTNFR1 the major TNFR controlling their maturation through the integrated activation of "canonical" NF-κB p65 and "non canonical" NF-κB RelB pathway[31,33]. Moreover, p65 NF-κB mediates the pro-survival effect mediated by TNFR1 in DC[31]. It is possible that DC of non-responding patients may also have increased survival and extended lifespan, as suggested by the induction of apoptosis detected during DC maturation in the presence of in vitro TNF blockade[28]. However, conflicting data have been observed in psoriasis skin[14,34]. In our in vitro system, adalimumab significantly decreased cell viability measured at day 8 of culture, with a minimal upward trend detected in non-responding patients. Similarly, although the overall number of CD11c+ DC was decreased at w12, it did not discriminate between PASI75 responders and non-responders, suggesting that, although affected by TNF-blockade, DC survival is unlikely to determine clinical response to adalimumab.

An intriguing possibility is that the expression of the immune checkpoint CD274/PDL-1 may be involved in underpinning DC as cellular determinants of response to adalimumab, as suggested by the correlation between CD274 expression and clinical response. Moreover, this effect may be genetically driven with at least one eQTL possibly explaining both CD274 expression and response to adalimumab. CD274, expressed by DC, is the main inhibitory ligand of CD279, also known as PD-1, expressed on T cells. Engagement of CD279 by CD274 alters T cell activity, by inhibiting T cell proliferation, survival and cytokine production[35]. CD279/PD1 is expressed on IL-17A+ T cells in psoriasis lesions and its blockade ameliorates inflammation in the aldara-induced psoriasiform skin inflammation model[36]. Another link between CD274/PDL-1 and psoriasis stems from the clinical observation than more than one third of cancer patients treated with Immune checkpoint inhibitors targeting cytotoxic T lymphocyte-associated antigen-4 (CTLA-4), CD279/PD-1 or CD274/PD-L1 develop cutaneous immune-related adverse events (irAEs), including characteristic psoriasis lesions[37].

Early studies have investigated the effect of TNF blockade on circulating 17A-producing T cells, with conflicting results. Two studies reported only a modest and insignificant decrease in frequency of circulating Th17 cells in patients receiving any of the anti-TNF agents[19,38], while another two found a significant decrease in circulating Th17 cells in patients undergoing adalimumab or etanercept treatment[39,40]. We found that the frequency of circulating Th17 cells did not change during adalimumab treatment, while circulating Tc17 cells dramatically decreased. More importantly, the frequency of IL17A+ T cells, either Th, Tc or Treg, was higher in PASI75 non-responders before commencing adalimumab, while Tc17 remained increased in PASI7 non-responders at week 12. An association between high baseline frequency of Th17 and lack of response to anti-TNFs has also been reported in RA[41]. In keeping with our blood data, the number of both Th17 and Tc17 cells in lesional psoriasis skin was significantly higher in PASI75 non-responders at w0, albeit disease severity was comparable in responders and non-responders. Moreover, only Tc17 cells remained higher in non-responders at w12. Thus, while the frequency of circulating IL-17+ T cells before the start of the therapy may have a role in determining the clinical outcome, an effective clinical response is only achieved in the presence of a significant decrease of Tc17 cells in both blood and skin. This is in line with the retainment of IL-17+ CD8+ Tissue-resident memory (T$_{rm}$) cells in clinically

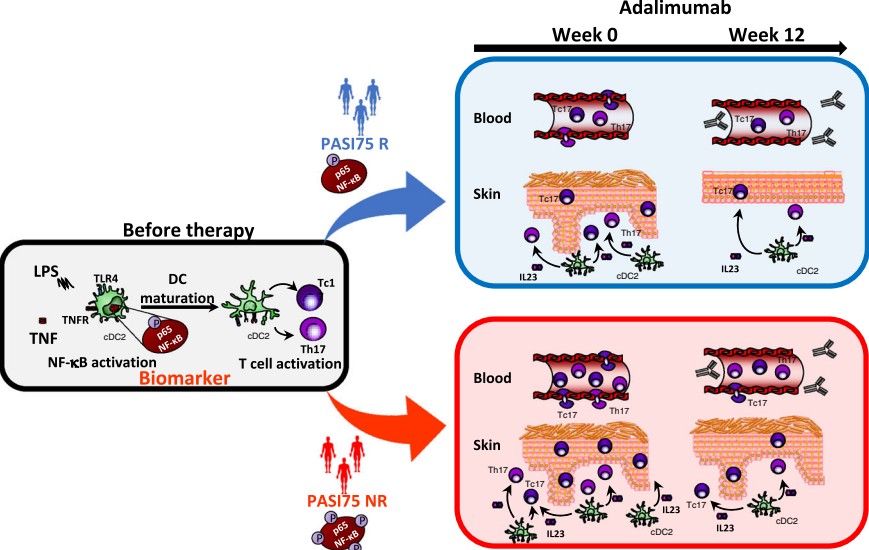

**Fig. 7 Mechanistic biomarker of response to adalimumab.** Phosphorylation of p65NF-κB in conventional Type 2 dendritic cells (cDC2) before therapy is an early predictive biomarker of response to adalimumab at week 12. NF-kB activation by TNF or TLR ligands, e.g., LPS, induces DC maturation which in turn leads to the activation of IL-17A producing T helper (Th)17 and CD8+ T cytotoxic (Tc)17 cells. This is enhanced in PASI 75 non responder patients (PASI75 NR), as shown in the cartoon by the increased number of phosphorylation sites on NF-kB underneath the patient silhouettes, and results in increased number of activated DC producing IL-23 in the skin and of Th17 and Tc17 in blood and skin before therapy, as compared to PASI75 responders (PASI75 R). TNF-blockade by adalimumab reduces DC and T cell responses but they remain elevated in the blood and skin of PASI75 non-responders underpinning the lack of clinical response.

resolved psoriasis lesion, where they are poised to be reactivated[42], and the beneficial effect of CD8 blockade in preventing psoriasis development in the AGR 129 xenotransplant model[21]. Tc17 could therefore play a role as a dynamic or surrogate biomarker, to monitor the effect of the drug at a clinical endpoint[11], rather than a predictive one. Nevertheless, our data show that despite the total inhibition of NF-kB signalling in T cells exerted by adalimumab, clinical response is more likely to be determined by its direct effect on DC, which in turn may affect T cells in blood and skin. This may also explain the stronger predictive value of LPS-induced NF-κB p65 phosphorylation in cDC2 when compared to the other blood immune traits which correlate with clinical response.

Taken together, our study uncovers key baseline determinants of response to adalimumab in psoriasis at molecular and cellular levels, from the more proximal signalling events, such as NF-κB activation in cDC2, to the ultimate effect on pathogenic IL-17+ T cells in blood and skin. Adalimumab is less effective in individuals in which increased NF-κB signalling induces a heightened maturation phenotype in DC and has increased frequency of IL-17+ T cells in blood and skin before the start of the therapy (Fig. 7). These findings suggest the existence of a specific disease endotype with direct implications on clinical response to adalimumab. Importantly, we identified, replicated and validated LPS-induced NF-κB p65 phosphorylation in cDC2 at baseline as a predictive biomarker of clinical response to adalimumab. Blood biomarkers are minimally invasive and transcription factor phosphorylation is already a validated diagnostic tool in cancer[43]. Our specific phospho-assay has already the key advantage to being used in cryopreserved cells, thus allowing maximal flexibility in terms of sample collection. Moreover, there is scope for further development into a more clinically scalable test (e.g. using whole blood) to be used for further clinical validation in prospective clinical trials. Importantly, such trials should also address the predictive value of the test not only for other anti-TNF agents but particularly for their more recent cheaper biosimilars[44], providing an alternative to the more expensive biologics targeting

the IL-23/IL-17 pathway. Finally, our findings have the potential to inform patient stratification in other immune-mediated inflammatory diseases such as inflammatory bowel disease and rheumatoid arthritis where TNF-blockade is also a therapeutic cornerstone[41,45].

## Methods

**Study design and patient cohorts.** The study included 67 adult patients with chronic plaque-type psoriasis with moderate-severe disease (PASI > 10) recruited into the Psoriasis Stratification to Optimise Relevant Therapy (PSORT) study[7] at six centers in the UK between May 2015 and July 2018 and due to start biologic therapy (ustekinumab or adalimumab) as part of routine clinical practice (Supplementary Data 1). Exclusion criteria were the use of systemic or biological therapy for psoriasis for 2 weeks prior to study entry, use of PUVA therapy for 3 months or UV-B for 1 month prior to study entry or use of topical treatments to site of biopsies (except for emollients) for 2 weeks prior to study entry, as well as serious/uncontrolled systemic disease or medical condition. From this study population we derived our discovery cohort of 16 patients receiving adalimumab (PSORT adalimumab discovery cohort) and 24 patients receiving ustekinumab (PSORT ustekinumab discovery cohort), all recruited in the Greater London area, and our replication cohort of further 27 patients receiving adalimumab (PSORT adalimumab replication cohort), recruited at various centers in the UK. All 43 PSORT patients receiving adalimumab are collectively referred as the PSORT adalimumab combined cohort (28 PASI75 responders, median baseline PASI = 13.5 ± 5.4; 15 PASI75 non-responders, median baseline PASI = 14.2 ± 5.4; Supplementary Fig. 1, Supplementary Data 1). Patients were largely of White European descent, both genders were represented (29 males/14 females), age range was 22–72 years, mean 45 years.

The independent clinical validation cohort (Supplementary Fig. 1) comprises of additional 15 psoriasis patients not receiving biological treatment at time of sampling. This included nine patients prospectively recruited into the BIODIP study[46] to receive adalimumab, sampled at baseline (w0) before commencing therapy, and six patients recruited into the Signature study[47], who had previously received adalimumab and had reached PASI75 at w12, but had then progressively lost clinical response 10–41 months after starting therapy and were retrospectively sampled 1–11 months after stopping adalimumab and before commencing secukinumab. Thus, for the purpose of our analysis they were considered PASI75 responders at w12.

This study was performed in accordance with the declaration of Helsinki and was approved by the London Bridge research ethics committee (REC numbers: 14/LO/1685; 11/LO/1692; 06/Q0704/18; 13/EE/0241). All participants provided written informed consent.

Full clinical and demographics patient information are in Supplementary Data 1.

Samples were randomly assigned to experiment batches, unless otherwise stated, and blinded for the person performing the experiments and the analyst carrying manual analysis steps such as cell gating. Some samples were excluded from the analysis following quality control (QC).

**Blood samples and PBMCs isolation**. Blood was collected at participating sites in BD Vacutainer Hemogard Closure Plastic K2-Edta Tubes (BD Biosciences, CA). For the discovery cohort, an aliquot of fresh blood was used for imaging flow cytometry (see below). Peripheral blood mononuclear cells (PBMCs) were isolated within 4 h of collection using Ficoll-Paque density gradient centrifugation in Leucosep tubes (Greiner Bio-One, Austria). Cells were viably cryopreserved in RPMI 1640 (Life Technologies, Carlsbad, Calif) supplemented with 11.25% Human Serum Albumin (Gemini Bio-Products, West Sacramento, Calif)/10% dimethyl sulfoxide (Sigma-Aldrich, St Louis, Mo) and stored in liquid nitrogen until used or shipped to St John's Institute of Dermatology. Prior to each experiment, cell count and viability were measured with Via1-Cassette on a NucleoCounter NC-200 (Chemometec, DK).

**Skin samples**. Six millimetres punch lesional psoriasis skin biopsy specimens were obtained from patients and bisected. Half was embedded in optimal cutting temperature compound (OCT; VWR, PA) and stored in liquid nitrogen until frozen-sectioned on a microtome-cryostat.

**Imaging flow cytometry**. Two hundred microlitres of fresh whole blood were incubated with Fc block (BioLegend, CA) for 10 min at room temperature (RT). Surface staining antibodies (Imaging flow cytometry panel, Supplementary Data 12 and Supplementary Fig. 2) were added and cells were stimulated with either TNF 20 ng/ml or LPS 1 μg/ml (Sigma-Aldrich, UK) or left unstimulated for 30 min at 37 °C. In some experiments, adalimumab 10 μg/ml (AbbVie, IL), which approximates adalimumab plasma concentration when given 40 mg biweekly[9], was added 30 min before stimulation. Erythrocytes were lysed and samples were simultaneously fixed with BD Phosflow Lyse/Fix Buffer (BD Biosciences, CA) for 10 min at 37 °C. Cells were spun down and intracellular staining primary rabbit NF-κB p65 (Cell Signalling, MA) (Imaging flow cytometry panel, Supplementary Data 12) was added in permeabilization buffer (0.1% Triton X in PBS) and incubated for 20 min at RT. Cells were washed in 2% BSA, 1 mM EDTA PBS and intracellular staining secondary donkey anti-rabbit IgG (BioLegend, CA)(Supplementary Data 12) was added in permeabilization buffer and incubated for 20 min at room temperature. Samples were washed and DAPI (Invitrogen, CA) 25 ng/ml was added prior to acquisition on an ImageStream MarkII imaging cytometer (Amnis, WA) at ×60 magnification. Gating of cell populations of interest and image analysis was performed using the IDEAS software (Amnis, WA). Cell populations for which <10 cells were acquired were excluded from the analysis. Nuclear translocation was assessed by the median internalization score feature for NF-κB signal in the nuclear mask based on the DAPI staining, which represents constitutive nuclear localization. The relative shift in the distribution between unstimulated versus stimulated sample was calculated using Fisher's Discriminant ratio ($R_d$ score) according to the formula:

$$Rd = \frac{|M_s - M_u|}{\sigma_s + \sigma_u} \qquad (1)$$

**Drug level measurements**. Blood samples were collected in Serum Sep Clot Activator Vacuette tubes (Greiner Bio-One, Austria) and centrifuged at $2000 \times g$ for 10 min, supernatant serum aliquots were obtained and frozen at –80 °C. Samples were shipped to Sanquin (Amsterdam, The Netherlands) and adalimumab concentration was measured by ELISA based on adalimumab's ability to bind TNF.

**Phospho flow cytometry**. Cryopreserved PBMCs were thawed in 22 batches of 1–2 patients per experiments, with each experiment comprising all time points (visits). Cells were rested overnight in RPMI-1640 medium supplemented with 10% heat-inactivated new-born calf serum (Life Technologies, CA) and penicillin/streptavidin 1%. On the next day, $2 \times 10^6$ cells were stimulated with either TNF 100 ng/ml or LPS 1 μg/ml (Sigma-Aldrich, UK) for 15 min at 37 °C. When a number of viable cells retrieved after resting was not sufficient, stimulation with LPS was prioritized. Samples were placed on ice and surface staining antibodies in brilliant stain buffer (BD Biosciences, CA) (Phospho Flow Cytometry panel, Supplementary Data 12 and Supplementary Fig. 4) were added for 30 min. Subsequent steps were carried out at RT. Cells were fixed with 1.5% paraformaldehyde in deionized water (Electron Microscopy Science, PA) for 10 min, permeabilized with Perm Buffer IV (BD Biosciences, NJ) for 20 min and stained with intracellular PE NF-κB p-p65 (Ser529) antibody (BD Biosciences, CA) for 1 h (Supplementary Data 12). Samples were acquired on a BD LSRFortessa. NF-kB p65 Ser529 phosphorylation was calculated as $\log_{10}$ mean fluorescence intensity (MFI) of stimulated sample /MFI of unstimulated sample, abbreviated as $\log_{10}$ fold change (FC) MFI. QC of the samples based on cell viability was carried out by building a ROC curve to identify viability cut-off which best discriminates good quality samples, pre-set as those with CD14$^+$ cell frequency >5% and LPS-induced NF-κB phosphorylation >$0.2\log_{10}$FC (Supplementary Fig. 16). As a result, samples with cell viability <93%

(sensitivity 90% and specificity 86%) were not included in the overall analysis. Inter-experiment variation in terms of cell frequency and level of NF-κB phosphorylation was assessed by running one internal control (IC) sample in each experiment. Inter-experiment variation for IC resulted to be minimal, while variability among patient samples was much larger, as expected for true biological variation (Supplementary Fig. 17).

**moDC generation, maturation and phenotyping**. Cryopreserved psoriasis patient PBMCs collected at w0 were thawed in five batches of four patients per experiment. Monocytes were isolated from PBMCs using MACS Separation Columns and CD14 MicroBeads (Miltenyi, Germany) as described by the manufacturer, and their purity was checked by flow cytometry (average purity >99%, Supplementary Fig. 18). Cells were cultured in RPMI-1640 medium supplemented with 10% heat-inactivated new-born calf serum (Life Technologies, CA), penicillin/streptavidin 1% for 6 days in the presence of IL-4 (500 U/ml) and GM-CSF (1000 U/ml) (PeproTech, UK) to obtain monocyte-derived dendritic cells (moDC). DC maturation was induced by adding LPS 250 ng/ml (Sigma-Aldrich, UK) in the presence or absence of adalimumab 10 μg/ml (AbbVie, IL) on day 6 of culture. On day 8, cells were stained with antibodies of the moDC panel (Supplementary Data 12) in brilliant stain buffer (BD Biosciences, CA) (Supplementary Table 13) for 30 min on ice, fixed with 1.5% paraformaldehyde (Electron Microscopy Science, PA) for 10 min and acquired on a BD FACSCanto. Samples with viability <85% were excluded from the analysis. Marker expression was reported as $\log_{10}$ MFI of LPS-matured moDC /MFI of immature moDC, abbreviated as $\log_{10}$ FC MFI. Experiments in healthy volunteers were run according to the same protocol by a different operator at a later time. To avoid operator-depended batch effects, a validation experiment was run with two samples each from psoriasis adalimumab PASI75 responders, PASI75 non-responders and HV and a normalization factor was calculated by dividing the median $\log_{10}$FC MFI of the psoriasis patient experiments and the healthy volunteer experiments by the median of these groups obtained in the validation experiment. Thus, normalized marker expression is reported in Supplementary Fig. 7 comparing patients and HV. Cytokine levels in day 8 moDC culture supernatant were assessed using Milliplex Map Human TH17 Magnetic Bead Panel (Merck Millipore, MA), following manufacturer's instructions.

**T cell phenotyping**. Cryopreserved PBMCs were thawed in 13 batches of 3–4 patients per experiments, with each experiment comprising w0 and w12 visits of interest. Cells were rested overnight in RPMI-1640 medium supplemented with 10% heat-inactivated new-born calf serum (Life Technologies, CA) and penicillin/streptavidin 1%. Next day $3 \times 10^6$ cells were stimulated with PMA (50 ng/ml, Sigma-Aldrich, UK) and ionomycin (1 mg/ml, Sigma-Aldrich, UK) in the presence of monensin (2 μM, Invitrogen, CA) and brefeldin A (3 μg/ml, Invitrogen, CA) for 3 h at 37 °C. Then, cells were centrifuged, resuspended in PBS and incubated with live/dead FVS780 reagent (BD Biosciences, CA) on ice for 30 min. Cells were spun down and stained with surface antibodies in brilliant stain buffer (BD Biosciences, CA) of the T cell phenotyping panel (Supplementary Data 12) for 30 min on ice, fixed and permeabilized with the Human FoxP3 Buffer Set (BD Biosciences, CA) following manufacturer's instructions. Intracellular antibodies were added for 1 h, cells were washed and acquired on a BD FACSCanto. Samples with viability <85% were excluded from the analysis. Inter-experiment variation in terms of cell frequency (Supplementary Fig. 19a) and cytokine production (Supplementary Fig. 19b) was assessed by running one IC sample in each experiment. Inter-experiment variation for IC resulted to be minimal, while variability among patient samples was significantly larger, as expected for true biological variation (Supplementary Fig. 19).

**Flow cytometry data acquisition, pre-processing and analysis**. Flow and phospho flow cytometry data were acquired using BD Standardized Application Setup, which allows to combine data acquired in different experiments thus minimizing batch-effects, and data were compensated using FACSDiva software. Signal anomalies derived from abrupt changes in the flow rate, instability of signal acquisition and/or margin events in the lower or upper limit of the dynamic range were filtered out of individual sample fcs files with the R package flowAI[48]. Cells were manually gated using Flowjo software 10.6.1. A graphic depicting the fluorescence distribution of all samples across all experiments for the different panels is shown in Supplementary Fig. 20. Representative gating strategy and cell hierarchy for each of the panel is shown in Supplementary Figs. 2, S4, S9, S18b and S21b. Positive gating for each fluorochrome parameter was established using fluorescence minus one (FMO) controls.

**DC phenotyping, data pre-processing and unsupervised clustering**. Cryopreserved PBMCs were thawed in 10 batches of 4–5 patients per experiment, with each experiment comprising all visits of interest. $2 \times 10^6$ cells were stained with live/dead FVS780 reagent (BD Biosciences, CA) for 15 min at room temperature, centrifuged and stained with antibodies of DC phenotyping panel (Supplementary Data 12) in Brilliant Stain buffer (BD Biosciences, CA) on ice. Subsequently, cells were fixed with 1.5% paraformaldehyde for 10 min, washed and acquired on a BD LSRFortessa using BD Standardized Application Setup. Samples with viability <85% were

excluded from further analysis. Inter-experiment variation in terms of cell frequency (Supplementary Fig. 21a) was assessed by running one IC sample in each experiment. Inter-experiment variation for IC resulted to be minimal, while variability among patient samples was significantly larger, as expected for true biological variation (Supplementary Fig. 21a).

Prior to further analysis,.fcs files were loaded into Flowjo, and HLA-DR+ cells were pre-gated in order to eliminate debris, doublets and dead cells (Supplementary Fig. 21b). Pre-gated.fcs files were imported into R and logicle transformation was applied using the flowCore package[49]. Subsequently, a multi-dimensional scaling (MDS) plot was created with R limma package including the expression of all markers, outlier samples with MSD1 < −0.15 were identified and excluded from further analysis (Supplementary Fig. 21c). Finally, cell populations were identified by unsupervised clustering using FlowSOM[22] and ConsensusClusterPlus[23] applied to all samples simultaneously. Briefly, a self-organizing map (SOM) was built using the logicle-transformed expression of the lineage markers (Lin, HLADR, CD11c, CD1c, CD141, CD123) with the BuildSOM function, where cells were assigned according to their similarities to 100 grid points of the SOM. Then, metaclustering of the SOM nodes was performed with the ConsensusClusterPlus function. Finally, some metaclusters were manually merged into the cell populations of interest according to their marker expression (Supplementary Fig. 21d, e).

**Skin immunofluorescence staining and quantification**. 6 μm-thick skin sections were fixed with 4% paraformaldehyde (Electron Microscopy Science, PA) and incubated with Image-iT FX Signal Enhancer (Invitrogen, CA) for 30 min, to block non-specific antibody binding sites. Next, sections were incubated at 4 °C overnight with primary antibodies (Skin immunofluorescent staining panels, Supplementary Data 12) in PBS buffer with 5% goat serum. Samples were washed with PBS-Tween 0.1% and incubated with secondary antibodies (Supplementary Data 12) diluted in PBS buffer with 5% goat serum for 1 h at RT. Subsequently, sections were washed with PBS-Tween 0.1% and streptavidin wash added for 30 min at RT. Finally, samples were washed, and nuclear staining was performed by embedding samples in Prolong Gold antifade reagent with DAPI (Invitrogen, CA). Stained sections were evaluated by epifluorescence in a Nikon DS-Qi2 sCMOS Eclipse Ti-2 inverted microscope (Nikon, JP) with a ×20 objective running NIS Elements. Image analysis was carried out with NIS Elements software. Briefly, an area of analysis of 750 × 750 μm was identified and a cellular mask created using bright spot detection with growing enabled in the DAPI image. A minimum intensity threshold was established for CD11c, CD4 or CD8 signals in order to identify the cells of interest. Then, either fluorescence intensity was measured in the cells of interest (CD83 and CD274) or a minimum intensity threshold was established to count the number of double-positive cells (IL23, IL17A). A region of interest (ROI) to limit the analysis to the dermis was added for the analysis of dermal DC.

**Statistical analysis**. For each experiment, data were subjected to quality control steps which may have resulted in some datapoints to be excluded from the overall statistical analysis for technical reasons. Specific exclusion criteria are described for each experimental methodology under the relative paragraph.

To study the relationship between immune biomarkers and the clinical response we modelled the latter first as a continuous variable by dividing PASI at week 12 by PASI at baseline (week 0) and expressing it as a percentage. This variable represents the % residual disease at week 12, e.g., a PASI75 responding patient has a residual disease of 25%.

Correlation between NF-κB nuclear translocation/phosphorylation, frequency of cytokine-producing T cell subset or CD274 expression, and residual disease were assessed using univariate linear regression in R for each combination of stimulus, cell type and timepoint. For imaging flow cytometry, when combinations of stimulation-cell type had a Rd score lower than 0.3 at baseline (i.e. there was no NF-κB translocation), they were was excluded from the correlation analysis.

To evaluate any potential confounding effect of adalimumab drug levels, a linear regression was performed for each stimulation, cell type and time point (with the exception of w0 where no drug had been administered yet) between NF-kB translocation and relative PASI, with the drug concentration (in μg/ml) measured at each visit added as a covariate.

For significant correlations, differences in NF-κB activation, marker expression or frequencies/counts of immune populations between PASI75 responders and non-responders (binary outcome) were assessed with the non-parametric Mann–Whitney $U$ test or the parametric unpaired $t$ test in R (both two-sided), as appropriate. Normality was assessed with the D'Agostino-Pearson test for groups with more than 8 values; if the group contained fewer than 8 values a non-normal distribution was assumed. The Benjamini–Hochberg approach (False Discovery Rate, FDR) was used to control for multiple tests, as appropriate. For comparisons between two-time points involving the same patient, differences were assessed with the non-parametric Wilcoxon test or the parametric paired t test, as appropriate. For comparisons between multiple time points involving the same patient, differences were assessed with the Kruskal–Wallis with Dunn's multiple comparisons post-test.

ROC curve analysis and area under the ROC curve (AUC) comparison were performed using the pROC R package[50]. For the long-term response analysis, patients that had switched treatment due to inefficacy were considered non-

responders. Predictive odds ratio was calculated using the sensitivity and specificity of the assay as reported in ref. [51].

$$\text{DOR} = \frac{\text{sensitivity} \times \text{specificity}}{(1 - \text{sensitivity}) \times (1 - \text{specificity})} \quad (2)$$

**Reporting summary**. Further information on research design is available in the Nature Research Reporting Summary linked to this article.

## Data availability

Whole blood eQTL summary statistics for CD274 gene were downloaded from www.gtexportal.org. All other data are provided in the article and its Supplementary files. Summary statistics for CD274 genotype in the PSORT genetic dataset and raw.fcs files can be obtained from the corresponding author upon reasonable request. Source data are provided with this paper.

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

## Acknowledgements

Supported by the PSORT Consortium, which is in turn funded by a Medical Research Council (MRC) Stratified Medicine award (MR/L011808/1). Partners of the PSORT consortium are AbbVie, the British Association of Dermatologists, Becton Dickinson and Company, Celgene Limited, GlaxoSmithKline, Guy's and St Thomas' NHS Foundation Trust, Eli Lilly, Janssen Research & Development, King's College London, LEO Pharma, MedImmune, Novartis Pharmaceuticals UK, Pfizer Italy, the Psoriasis Association, Qiagen Manchester, Queen Mary University of London, the Royal College of Physicians, Sanquin Blood Supply Foundation, the University of Liverpool, the University of Manchester, and Newcastle University. We particularly acknowledge generous in-kind support from the PSORT industrial partner Becton Dickson. All decisions concerning analysis, interpretation, and publication are made independently of any industrial contribution. This research was also supported by the National Institute for Health Research (NIHR) Biomedical Research Centre based at Guy's and St Thomas' NHS Foundation Trust and King's College London, the Newcastle NIHR Biomedical Research Centre, and the NIHR Manchester Biomedical Research Centre. The views expressed are those of the author(s) and not necessarily those of the NHS, the NIHR, or the Department of Health and Social Care. The GTEx Project was supported by the Common Fund of the Office of the Director of the National Institutes of Health, and by NCI, NHGRI, NHLBI, NIDA, NIMH, and NINDS. ND is supported by Health Data Research UK (MR/S003126/1). NJR is supported by the Newcastle NIHR Biomedical Research Centre and the Newcastle NIHR Medtech and In vitro diagnostics Co-operative. CEMG and NJR are NIHR Senior Investigators. We are grateful to psoriasis patients and healthy volunteers for their participation. We acknowledge the enthusiastic collaboration of the dermatologists and specialist nurses in the UK who recruited to this study, in particular, Prof. David Burden (Western Infirmary, Glasgow), Dr Evmorfia Ladoyanni (Russells Hall Hospital, The Dudley Group NHS Foundation Trust, Dudley), Dr Richard Parslew (Royal Liverpool& Broadgreen University Hospital NHS Trust), and Dr Gayathri Perera (West Middlesex University Hospital, Chelsea and Westminster Hospital NHS Foundation Trust). We thank Theo Rispens and Annick de Vries at Sanquin, for measuring adalimumab drug levels. We thank Alice Russel, Michael Duckworth, Tejus Dasandi, Nadya Dinev, Freya Meynell (London), Tom Ewen, and Dhanisha Lukka (Newcastle) for sample and data management, and Federica Villanova (London) for her contribution to obtain ethical approval. We thank Esme Nichols (Newcastle) for skin sectioning, Susanne Heck, Anna Rose and PJ Chana at the BRC Flow Cytometry Platform at NIHR Guy's and St Thomas' Biomedical Research Centre and Virginia Silio and Isma Ali at the Nikon Imaging Centre at Kings College London for technical assistance. We thank Ruth Williams at Novartis for help with clinical data of patients recruited into the Signature study. We thank Dr Brigitta Stockinger for her critical reading of the manuscript.

## Author contributions

R.A.E. performed experiments, analysed and interpreted data and drafted the manuscript. H.A. and K.G. processed samples and performed experiments. I.T. and Z.C. processed and managed samples. S.S. performed experiments. C.A., J.S., N.D. and E.D.R. analysed data and provided statistical advice. H.S. provided nursing support. A.C. provided clinical samples. F.O.N. conceived and designed the PSORT program and acquired funding. M.R.B. acquired funding and contributed to the design and implementation of the PSORT program. C.E.M.G., J.N.B., N.J.R., R.B.W. and C.H.S. conceived and designed the PSORT program, acquired funding, supervised the clinical study and sample collection, and provided insight into data interpretation. The PSORT consortium designed the clinical study, secured funding, collected psoriasis patient samples and clinical data. P.D.M. interpreted data, designed and supervised the study, and wrote the manuscript. All the authors reviewed the data and contributed to the manuscript.

## Competing interests

The authors declare the following competing interests: C.A. is an employee of DIGNOSIS Ltd. E.d.R. and F.O.N. are currently employees of Sanofi. M.R.B. has received honoraria and/or research grants from Janssen, Servier and Lilly. R.B.W. has received honoraria and/or research grants from AbbVie, Almirall, Amgen, Boehringer Ingelheim, Celgene, Janssen, Leo, Lilly, Novartis, Pfizer, Sanofi, Xenoport, and UCB. N.J.R. has received research grants from GSK-Stiefel and Novartis; and other income to Newcastle University from Almirall, Amgen, Janssen, Novartis, Sanofi Genzyme Regeneron and UCB Pharma Ltd for lectures/attendance at advisory boards. C.E.M.G. reports grants and/or personal fees from AbbVie, Almirall, Amgen, BMS, Celgene, Galderma, Janssen, Leo Pharma, Lilly, Novartis, Sandoz and UCB Pharma. J.N.B. has received honoraria and/or research grants from AbbVie, Almirall, Amgen, Boehringer-Ingelheim, Bristol Myers Squibb, Celgene, Janssen, Leo, Lilly, Novartis, Samsung, Sun Pharma. C.H.S. has received departmental research funding from AbbVie, GSK, Pfizer, Novartis, Regeneron, and Roche. P.D.M. reports grants and/or personal fees from Janssen, Novartis, and UCB Pharma. All the other authors declare no competing interests.

### Additional information

### the PSORT Consortium

Christopher E. M. Griffiths[7,13], Jonathan N. Barker[1,2,13], Michael R. Barnes [6], Paola Di Meglio[1,2], Richard Emsley[9], Andrea Evans[7], Katherine Payne[10], Nick J. Reynolds[8,13], Catherine H. Smith[1,2,13], Deborah Stocken[11] & Richard B. Warren[7,13]

[9]Department of Biostatistics and Health Informatics, Institute of Psychiatry, Psychology & Neuroscience, King's College London, London, UK. [10]Manchester Centre for Health Economics, Division of Population Health, Health Services Research and Primary Care, The University of Manchester, Manchester, UK. [11]Leeds Institute of Clinical Trials Research, University of Leeds, Leeds, UK.

