## [Peer Review File · Nature Communications]

REVIEWER COMMENTS

Reviewer #1 (Remarks to the Author):

Andres-Ejarque et al present a manuscript analysing the correlations between the responsiveness of adalimumab (anti-TNF antibody) and NfKB activation in blood and skin immune cells. The main finding of the manuscript, i.e. the predictive power of NfKB status in DC2 pre – therapy is novel and exciting for the clinical application in therapy of skin disease, allowing better stratification of patients with a relatively simple assay.

The bulk of analysis have been done on patients with psoriasis, on treatment with different biologics, which provides strength to the clinical applicability.

The authors provide a very comprehensive description of the system, and measure many aspects of the immune activation prior and post therapy. However, despite claiming to provide mechanistic explanation for the observed associations, the evidence for molecular mechanisms has not been provided.

Major points:

1) Mechanisms of action

While authors postulate stimulation with LPS provides the best indicator for patient responsiveness to therapy with adalimumab, the link between LPS and the anti-TNF agent has not been explained. It is not clear, why authors would choose stimulation with LPS as a biomarker in the first place, or how it is related to psoriasis. In fact, in the cited paper (Lehner, M. et al. Autocrine TNF is critical for the survival of human dendritic cells by regulating BAK, BCL-2, and FLIPL. J Immunol 188, 4810-4818, doi:10.4049/jimmunol.1101610 (2012).) LPS is NOT inducing autocrine TNF, it was R848 which prevented apoptosis in the lack of sufficient TNF α in LPS treated DC. In fact, TNF signalling constitutes much weaker biomarker, and is far less predictive for the response to adalimumab.

Thus, the results suggest, that the adalimumab blocks the NfKB action independently of TNF α signalling, and a mechanistic explanation for this should be sought, perhaps via whole transcriptome/proteome investigation of the effect of adalimumab on DCs or perturbation assays, revealing adalimumab mechanism of action on DC2. It would also be important to compare the predictive power to another anti-TNF antibody, e.g. etanercept

2) Use of discovery cohort vs validation cohort

Analysis of patients from different treatment strategies is very interesting. However, it is unclear, why authors decided to split the adalimumab cohort into two, i.e. discovery and validation, as they are not used for any train-test analysis, and class discovery in the test/validation cohort. It would be very exciting to test the power of the prediction in a separate, dedicated cohort of patients.

3) Adalimumab blocks LPS induced maturation

In the MoDC experiment, Figure 3, although the results are not statistically tested, it seems that the anti-TNF treatment reduces activation by LPS, both in the responders and, to a lesser degree, in non-responder. Similarly to point 1, a mechanistic explanation of adalimumab action is needed

4) Up-regulation of CD83, CD40 and CD86 in DC induced by LPS is so canonical, it cannot really be used as a novel mechanical explanation, it is simply confirming the NfKB stimulation

Minor comments:

Fig 1A – describe FACS plot on the figure

FIG 1B – add unstimulated + add quantification

1C – what do red and blue squares mean, number of patient testes need to be included in the figure legend

1D – should there be a different curve fitted? 13 dots on the Adalimumab, 16 patients, why?

1G – The numbers need to explained, for NR only 4 patients, and all show a degree of reduction, lack of statistical significance might be due to patient number, but additionally, it does not fit graph from the same cohort in the supplementary figure

Supplementary figure S3A – legend confusing – A -which patients? Is it in Adalimumab treated patients?. B, C – specify patient cohort, E: why so many non-responders in here compared to the main figure?

Figure 3C – looks like adalimumab blocks the maturation in non-responders, this should be referred to in the text and explained

Text: state the major skin population, 4th line

Why use two PASI scores for the binary model?
List all 11 cell subsets

The wording needs to be adjusted, it reads as the residual disease was observed in DC.
“In keeping with the correlation between NF- κ B nuclear translocation and residual disease observed in DCs (Figure 1d), ““Notably, however, we found a statistically significant correlation ($r^2=0.58$, $p=0.002$, $FDR < 0.05$) between LPS-induced NF- κ B translocation at w0 in DCs and % residual disease, with increased NF- κ B activation in patients with higher residual disease at w12”

Why do you say “analytically validating”? previous assay was based on similar principles, analytically is not needed, and incorrect in this sentence

“The NF- κ B activation cascade includes phosphorylation of p65 NF- κ B subunit, such as at Ser529, which regulate NF- κ B function 17. Thus, we developed a 13-colours phospho-flow cytometry panel”, These two sentences do not have a causal link, more explanation about the need for 13 color panel is needed.

Correlation at $r^2= 0.24$ is not really strong, it is better to confirm only LPS was predictive

It is not clear how many comparisons were done in the validation cohort – was the second one focused only in DC compartment?

It would be easier to read if R and NR were replaced with words

Why Th17 did not decrease? Up-regulation of CD40, CD86 and CD83 does not explain Tc17 specific stimulation

We observed a similar trend in our adalimumab cohort for CD274 protein expression in DC subsets at various time points – it needs explaining, how did you measure a similar trend to qQTL w/o measuring SNP?

Discussion – why the effect is specific to DC2?

The manuscript needs editing to improve the flow, and to remove the unclear/grammatically incorrect sentences.

Reviewer #2 (Remarks to the Author):

This study represents an excellent example of deep phenotypical analysis of a psoriasis patient cohort which aims to address a significant unmet need in terms of identifying novel biomarkers which may offer utility in predicting responsiveness to adalimumab. Although the assay identified may be too cumbersome, in its present format, for routine use in the clinic, the study is novel, comprehensive and experimentally sound. It will be of significant interest to both dermatologist clinicians and researchers involved in studying the pathogenesis of psoriasis.

Some specific comments include:

1. There is a discrepancy and lack of consistency in patient numbers as described in the text versus those analysed in the figures For example see figure 1 & 2, (16 in cohort versus 13 in 1D

and 12 in 1G). If some patients have been excluded from the analysis the authors should be far more explicit for their reasons for exclusion than currently outlined in the relevant methods sections as this may have significant impact on results presented.

In figure 1G, it looks as though there is a trend towards decreased Rd score in the NR patients also but this may be statistically NS due to low numbers. What is the P value here? See point about sample exclusion criteria above.

2. For the blood samples used in figure 1, was any analysis of relative levels of adalimumab present in patients blood undertaken post treatment undertaken? It would be interesting to determine whether individual variations in levels of circulating Ab among patients could have any impact on the observations made. This is important as this early data is framed as introducing the subsequent analysis of DC subsets.

3. The observations concerning the relative levels of Tc17 cells and their possible predictive role in determining anti-TNF responsiveness are intriguing. However data from the ROC curve analysis in Supp. Fig 15 indicate that Tc17 levels at week 0 are not as good accurate predictor of response. These data should be discussed in more detail.

Minor comments:

Typos throughout the manuscript eg. line 220, line 538

Representative FACS plots in Figs 1A & 2A should be labelled

Reviewer #3 (Remarks to the Author):

Andres-Ejarque and colleagues report that the activation potential of dendritic cells, especially cDC2 cells, at baseline correlates with the clinical outcome of adalimumab therapy at week 12 in psoriasis patients. The study is part of a large consortium aiming at identifying psoriasis biomarkers. Such biomarkers of response are a high unmet medical need. The manuscript is well written and of high interest to the broad readership of Nature Communications. The complex nature of the biomarker somewhat dampens the enthusiasm of clinical utility, but the degree of innovation is very high and the manuscript should be published after some corrections.

1.) Statistical analysis: the ratio of n number in experiments and number of readouts requires a Bonferroni-correction. How do corrected p values look like? There is a risk of over-interpretation.
2.) Influence of skin severity – many findings in the manuscript point towards the possibility that non-responders had a more severe inflammation at baseline than responders, and this is reflected by the activation potential of DCs. In other words, does the level of NF-kB translocation merely reflect disease severity and the more severely affected patients respond less well to adalimumab? Also results described in lines 247/248 and the higher number of IL-17+ cells in non-responders shown in figure 5 would argue in that direction. The authors address that in supplemental table 5, which is not very well explained, and only for the cDC2 experiments in figure 2. It would help to clarify here.

3.) Overinterpretation: results table 2 and figure 1c: please modify sentence lines 195/196, as we would always expect a delayed clinical response to molecular changes at cellular level. Similarly, Lines 299/300 – why do the authors refer to cDC2 if moDC were investigated? Finally, the statement in lines 304-306 is not supported by data – there is no mechanistic link established in the work between NF-kB activation potential of DC subsets and the number of IL-17+ T cells.

4.) Statistics: Line 412/413: was the cut-off defined before the validation cohort was analysed? If not, there is a risk of overfitting

5.) Figure 1: relative PASI calculation, the remaining PASI – is it true that all patients improved by at least 50% after week 12 under Adalimumab, and how could the different picture in the discovery cohort (figure 2) influence the findings?

We thank the Reviewers for the comprehensive and insightful review of our work. We have addressed all the points raised in our point-by-point reply below. We believe these comments and queries have significantly improved the manuscript.

All changes are highlighted in yellow in the revised manuscript. Pages and lines mentioned in the replies below refer to the actual pages and lines in the revised manuscript.

REVIEWERS' COMMENTS

Reviewer #1 (Remarks to the Author):

Andres-Ejarque et al present a manuscript analysing the correlations between the responsiveness of adalimumab (anti-TNF antibody) and NfκB activation in blood and skin immune cells. The main finding of the manuscript, i.e. the predictive power of NfκB status in DC2 pre – therapy is novel and exciting for the clinical application in therapy of skin disease, allowing better stratification of patients with a relatively simple assay.

The bulk of analysis have been done on patients with psoriasis, on treatment with different biologics, which provides strength to the clinical applicability.

The authors provide a very comprehensive description of the system, and measure many aspects of the immune activation prior and post therapy. However, despite claiming to provide mechanistic explanation for the observed associations, the evidence for molecular mechanisms has not been provided.

We thank Reviewer 1 for their positive assessment of our work and address their comments and concerns below.

Major points:

1) Mechanisms of action

While authors postulate stimulation with LPS provides the best indicator for patient responsiveness to therapy with adalimumab, the link between LPS and the anti-TNF agent has not been explained. It is not clear, why authors would choose stimulation with LPS as a biomarker in the first place, or how it is related to psoriasis.

We initially included LPS in our imaging flow cytometry experiment monitoring NF-κB nuclear translocation before and during adalimumab therapy, as LPS stimulation is a well-known inducer of NF-κB activation. We have now clarified why we have used LPS on page 8, lines 11-12. We agree with Reviewer 1 that the link between LPS and psoriasis is not immediately apparent; LPS is used here as model stimulus for NF-κB. We now speculate in the Discussion that the strongest predictive value of *in vitro* LPS vs TNF stimulation may be due to the stronger NF-κB activation and DC maturation induced by LPS (page 23, lines 15-16; page 25, line 23; page 26, lines 1-2). While there may have been an initial element of serendipity in our findings using LPS, the statistically significant correlation between LPS-induced NF-κB nuclear translocation in DC and lack of clinical response to adalimumab (**Figure 1d** and **Supplementary**

Table 2) has been refined, identifying cDC2 as the implicated subset, and replicated ($r^2 = 0.57$, $FDR = 10^{-4}$) using NF- κ B phosphorylation as alternative read-out (**Figure 2**).

In fact, in the cited paper (Lehner, M. et al. Autocrine TNF is critical for the survival of human dendritic cells by regulating BAK, BCL-2, and FLIPL. J Immunol 188, 4810-4818, doi:10.4049/jimmunol.1101610 (2012).) LPS is NOT inducing autocrine TNF, it was R848 which prevented apoptosis in the lack of sufficient TNF α in LPS treated DC.

We apologize for the wrong citation, due to an unfortunate mix-up with the Endnote reference software. The article we intended to cite was Stamatatos et al. "LPS-induced cytokine production in human dendritic cells is regulated by sialidase activity" *J Leukoc Biol* 2010, in which the authors show production of TNF induced by LPS in human dendritic cells. The reference, #32 in the manuscript, has been changed accordingly on page 23, line 17.

Nevertheless, we can only speculate on the potential link between LPS and TNF and have revised the manuscript accordingly (page 23, line 17).

In fact, TNF signalling constitutes much weaker biomarker, and is far less predictive for the response to adalimumab.

We concur with Reviewer 1 that TNF-induced NF- κ B phosphorylation is a much weaker biomarker and far less predictive of response to adalimumab. As discussed above, we hypothesize that this may be due to the stronger NF- κ B stimulation induced by LPS vs TNF in cDC2 (**Supplemental Fig 5**), in line with the weaker DC maturation ability exerted in vitro by TNF vs LPS reported in the literature (page 23, line 16). Nevertheless, the fact that NF- κ B phosphorylation in cDC2 induced by either LPS or TNF correlates to some extent with clinical response to adalimumab suggest that adalimumab non-responders have an enhanced intrinsic propensity to activate the NF- κ B pathway, regardless of the type of stimulus. In support of this hypothesis, we have preliminary data which we would like to make the Reviewer aware of. Using IL- β as an additional NF- κ B inducer in baseline samples, we have detected a putative correlation ($r^2 = 0.2$, $p = 0.02$, $FDR = 0.17$, $n = 25$, data not shown) between IL- β -induced NF- κ B phosphorylation in cDC2 at baseline and clinical response at W12. Thus, we believe that, notwithstanding the minor biomarker potential of stimulations other than LPS, the correlations detected with TNF (and IL-1 β) lend further support to the existence of an enhanced intrinsic propensity to activate the NF- κ B pathway in adalimumab non-responders which makes them more refractory to the clinical effect of adalimumab (see also below).

Thus, the results suggest, that the adalimumab blocks the NfKB action independently of TNF α signalling, and a mechanistic explanation for this should be sought, perhaps via whole transcriptome/proteome investigation of the effect of adalimumab on DCs or perturbation assays, revealing adalimumab mechanism of action on DC2.

The ultimate finding of our study is that clinical response to adalimumab in psoriasis correlates with, and can be predicted by, NF- κ B activation in cDC2 at baseline, with LPS as model stimulus with the best predictive value (**Fig 1d**, **Fig.2e**, **Fig. 6a**, **Fig. 6d**). Moreover, we provide some novel insights into adalimumab's mechanism of action, showing that: i) the full inhibition of TNF signalling in T cells detected in whole blood during therapy (**Fig 1c**) does not determine nor correlate with clinical response (**Supplemental Table 2**); ii) adalimumab only partially inhibits TNF signalling in DC in whole blood (**Fig. 1c**), with significant reduction at week 12 of TNF-induced NF- κ B nuclear translocation in DC of adalimumab responders but

not of non-responders (**Fig. 1g**); iii) the increased propensity to activate NF- κ B in adalimumab non-responders is accompanied by increased DC maturation (in vitro and in the skin, **Fig. 3bc** and **Fig5bc**) as well as increased T17 responses (in blood and skin, **Fig. 4bc** and **Fig. 5ef**). Nevertheless, we acknowledge that, while our data identify DC as the cellular target of adalimumab in psoriasis and describe the downstream functional consequences in blood and skin, they do not actually explain what causes the intrinsic greater propensity of DC of non-responders to activation and maturation. We agree with the Reviewer that further work is needed to fully elucidate the molecular mechanisms underpinning the enhanced propensity of cDC2 of non-responders to activate the NF- κ B pathway. The experiments she/he suggests would be undoubtedly very informative for that purpose and we are grateful for the suggestion. Nevertheless, we believe that these broad and high-dimensional experiments, which will inevitably open new lines of investigations, are more suited to a separate, subsequent study and outside the scope of the current manuscript which is aimed at identifying immune biomarkers predictive of clinical response to adalimumab. Nevertheless, in response to the Reviewer's comment we have now removed any claim of having identified the underlying mechanism of the biomarker, both in the abstract (page 4, line 10) and in several places in the text (e.g. page 7, line 12; page 8, line 6; page 14, line 2; page 21, lines 8-9; and only suggest the existence of potential mechanistic links (e.g. page 23, line 13 and line 17).

It would also be important to compare the predictive power to another anti-TNF antibody, e.g. etanercept

We agree with the Reviewer that it will be important to assess the predictive value of the identified biomarker in patients treated with other anti-TNF biologics, e.g. etanercept.

However, adalimumab remains the main anti-TNF used in clinical practice in psoriasis (as well as across other immune mediated diseases) based on superior effectiveness (to etanercept), safety profile and now, with the advent of adalimumab biosimilars, low drug acquisition costs

Thus, the limited number of patients currently commencing another anti-TNF therapy poses a practical challenge to carry out this analysis within a reasonable timeframe. We are nevertheless committed to answer this important question and will be collecting clinical samples of patients commencing etanercept, or the more recently approved certolizumab, as they become available for a future, dedicated study.

2) Use of discovery cohort vs validation cohort

Analysis of patients from different treatment strategies is very interesting. However, it is unclear, why authors decided to split the adalimumab cohort into two, i.e. discovery and validation, as they are not used for any train-test analysis, and class discovery in the test/validation cohort.

We decided to split the PSORT adalimumab cohort into Discovery and Replication early on during patient recruitment, due to the methodological requirements of the imaging flow cytometry assay to detect NF- κ B nuclear translocation, which had to be performed in fresh blood. This limited our PSORT adalimumab discovery cohort to patients recruited within the Greater London area, from where it was possible to courier the fresh blood to our lab at Guy's Hospital in London and perform experiments on the same day. Thus, patients recruited at other centres in England, alongside other patients recruited in Greater London at a later date,

became our PSORT adalimumab replication cohort. This also meant that to replicate and refine our early imaging flow cytometry findings in a larger patient cohort we had to use a related, yet different, analytical read-out, assessing NF-kB phosphorylation, which could be robustly performed in frozen PBMCs, rather than nuclear translocation. To validate this different assay, we first evaluated NF-kB phosphorylation in the same PSORT adalimumab discovery cohort in which we had assessed NF-kB nuclear translocation confirming that: i) LPS-induced NF-kB phosphorylation in cDC2 at baseline correlated with clinical response (**Fig 2b**); ii) the LPS-induced NF-kB translocation signal detected by imaging flow cytometry in DC at baseline highly correlated with the LPS-induced NF-kB phosphorylation detected by phosphoflow in cDC2 at baseline in the same patients (**Fig 2d**). Therefore, we assessed NF-kB phosphorylation in the PSORT adalimumab replication cohort (**Supplementary Fig 6B**) and performed the analysis in the discovery + replication= PSORT adalimumab combined cohort (**Fig. 2e**). Finally, we used an additional, independent (non-PSORT) clinical validation cohort of adalimumab patients for the actual validation step, classifying new patients (**Fig. 6e**) using the phosphorylation cut off previously identified in the PSORT adalimumab combined cohort (**Fig.6a**). We have added further details about the relationship between the PSORT cohorts in **Results** (page 11, lines 10-12; page 12, line 12; page 20, lines 6-8) and **Material and Methods** (page 27, lines 12-18).

It would be very exciting to test the power of the prediction in a separate, dedicated cohort of patients.

We completely agree with the Reviewer and respectfully note that we have indeed assessed the predictive power of LPS-induced NF-kB phosphorylation in cDC2 at baseline in a separate and independent cohort of 15 patients receiving adalimumab (clinical validation cohort, **Fig. 6e** and **Supplemental Figure1**). The distinction of this independent clinical validation cohort from the PSORT discovery and replication cohorts has been further clarified in **Results** (page 20, lines 6-8).

3) Adalimumab blocks LPS induced maturation

In the MoDC experiment, Figure 3, although the results are not statistically tested, it seems that the anti-TNF treatment reduces activation by LPS, both in the responders and, to a lesser degree, in non-responder. Similarly to point 1, a mechanistic explanation of adalimumab action is needed

We thank the reviewer for raising an important point here. In **Fig 3c**, we originally compared the expression levels of the maturation molecules CD54, CD80, CD83 and CD86 induced *in vitro* by LPS, in responders and non-responders, either in presence or in absence of adalimumab. We found that maturation markers were consistently increased in non-responders vs responders, both in the presence and absence of TNF blockade. Following the Reviewer's comment, we have now statistically tested the specific effect of *in vitro* adalimumab in paired samples of responders and of non-responders and show the results of the new statistical analysis in **Fig 3c**, using hashtags (#) instead of asterisks (*) to denote p values, in order to differentiate between the two analyses, as the original one is performed in unpaired samples.

In keeping with the significant inhibition by adalimumab of LPS-induced DC maturation in the overall cohort, i.e. without discriminating between responders and non-responders shown in **Supplemental Figure 7b**, we found that the drug was able to significantly reduce

the expression of maturation markers in both responders and non-responders, albeit with some differences for the different markers. Thus, while adalimumab overall inhibits DC maturation, this effect is not sufficient to offset the increased maturation propensity displayed by non-responder patients, which consistently display increased levels of maturation markers than responders, both in the presence and absence of the drug. We have added and discussed the results of the additional analysis on page 14, line 20, page 15, lines 3-4, and page 23, lines 4-5.

4) *Up-regulation of CD83, CD40 and CD86 in DC induced by LPS is so canonical, it cannot really be used as a novel mechanical explanation, it is simply confirming the NfKB stimulation.*

We definitely agree with the Reviewer that the notion of LPS-induced up-regulation of CD83, CD40 and CD86 in DC is well-established. Indeed, we prefaced our *in vitro* maturation experiments with this background knowledge and have further clarified this in the revised manuscript (page 13, line 23). Having observed an increased propensity for NF- κ B activation in DC of non-responders before anti-TNF treatment (**Figure 1** and **Figure 2**), we hypothesized that lack of clinical response to adalimumab may therefore be linked to an increased propensity to upregulate the expression of co-stimulatory molecules. As referenced in the discussion (page 22, lines 11-13. Ref 16: Zaba et al. 2007. Ref 28: Baldwin et al 2010), TNF blockade had been previously shown to impair maturation of DC generated *in vitro* from healthy volunteers and rheumatoid arthritis patients, resulting in reduced levels of HLA-DR and co-stimulatory molecules and to diminish the expression of co-stimulatory molecules in psoriasis skin DCs. However, this effect had not been studied before in responder and non-responders. Our data build on these previous findings, validating DCs as the cellular target of adalimumab and highlighting their intrinsic maturation propensity and status as the key factor linked to clinical outcome in psoriasis patients treated with adalimumab.

Minor comments:

1) *Fig 1A – describe FACS plot on the figure*

Figure 1a is a cartoon depicting the experimental workflow, in which we have included some representative images to show the identification of cell population and their fluorescent images via imaging flow cytometry. We have better described the figure in the revised legend (page 45). Moreover, we have labelled the axes of the representative flow cytometry dot plot.

2) *FIG 1B – add unstimulated + add quantification*

Figure 1b depicts representative fluorescent images of three selected cell populations within PBMCs to show the nuclear localization of NF- κ B following stimulation with TNF or LPS, before and during adalimumab therapy. We have intentionally not included the unstimulated cells, which being different for each of the 4 time points would nearly double the size of the panel, while not substantially enhancing its message. Nevertheless, we show representative unstimulated cell images for all seven cell populations studied in **Figure S2b** and provide the quantification of the unstimulated nuclear localization in **Figure S3a**.

3) *1C – what do red and blue squares mean, number of patient testes need to be included in the figure legend*

The different coloured frames represent the different levels of FDR significance as already indicated in the **Figure legend** (page 45, lines 22-23). The number of patients has been added to the legend (page 45, line 13).

4) 1D – should there be a different curve fitted? 13 dots on the Adalimumab, 16 patients, why?

With regards to the type of curve fitted, we acknowledge that a different fit could be an option for the data shown in figure 1D. However, as we tested the correlation with response for different cell types, stimulations and time points, we chose to fit a simpler and more generalizable linear regression model, to consistently encompass all the conditions tested. We thank the reviewer for raising the important point regarding the apparent discrepancy between the 16 patients composing the adalimumab discovery cohort and the 13 dots present in Figure 1d. This is due to the quality control (QC) step we have implemented in our analytical pipeline to ensure data included in the final statistical analysis were robust. QC criteria were pre-determined for each experiment and were stated in **Material and Methods** under the respective headings. For NF-κB nuclear translocation, we excluded datapoints for cell type/timepoint/stimulation for which we could not acquire at least 10 cell/cell type, as stated in **Material and Methods** (page 30, lines 3-4). Moreover, combinations of stimulation-cell type with a Rd score lower than 0.3 at baseline (i.e. no NF-κB translocation had occurred) were not included in the correlation analysis (page 37, lines 5-7). The latter information was previously reported when describing the imaging-flow cytometry methodology and has now been more appropriately moved when describing the correlation analysis. To further clarify the existence of data exclusion criteria for each experiment that may give rise to apparent discrepancies in patient numbers, we have re-stated their existence under the revised **Statistical analysis** paragraph, in the revised **Material and Methods** (page 36, lines 17-19). Finally, we now specify in the revised **Supplemental table 10** which samples passed the QC step and were included in the analysis for each experiment. Please see also replies to *major point 1* raised by Reviewer 2 and *major point 5* raised by Reviewer 3.

5) 1G – The numbers need to explained, for NR only 4 patients, and all show a degree of reduction, lack of statistical significance might be due to patient number, but additionally, it does not fit graph from the same cohort in the supplementary figure

In **Figure 1g** we show TNF-induced NF-κB nuclear translocation in DCs at baseline and w12 in PASI75 responders and non-responders, while **Supplemental Figure 3e** shows LPS -induced NF-κB nuclear translocation in DCs in PASI90 responders and non-responders, hence the higher number of non-responders. We appreciate the Reviewer's comment about the limited number of PASI75 non-responders and have revised the text to: *i)* acknowledge the downward trend observed in non-responders; *ii)* include the exact p value ($p=0.12$) observed in non-responders (Page 10, line 22). Please also see reply above to *point 4* and reply to *major point 1* raised by Reviewer 2, in which we also address the apparent discrepancy between the number of PASI75 non responders in **Fig 1d** and **Fig 1g**, due to data QC.

6) Supplementary figure S3A – legend confusing – A -which patients? Is it in Adalimumab treated patients? B, C – specify patient cohort, E: why so many non-responders in here compared to the main figure?

We are sorry for the lack of details in Supplementary figure S3. Additional details have been added to the figure legend (Page 6 Supplemental File) . Panel A and B refer to adalimumab-treated patients, while C refers to ustekinumab-treated patients.

Supp figure 3e refers to adalimumab PASI90 responders vs non-responders whereas figure 1F refers to adalimumab PASI75 responders vs non-responders. As expected, less patients achieved PASI90 than PASI75, thus explaining a greater number of non-responders in **Supplemental Figure 3e**.

7) Figure 3C – looks like adalimumab blocks the maturation in non-responders, this should be referred to in the text and explained.

This has been raised previously by this Reviewer under major point 3. Please see our reply above.

8)Text: state the major skin population, 4rth line

We aren't exactly sure to which part of the manuscript this comment refers to. We have carefully checked the 4th line of each page on the submitted pdf and hypothesized that the comment may refer to line 4 on page 22 of the first submission (line 449 on pdf: "DC are highly implicated in psoriasis pathogenesis"). Thus, we have stated the major skin population implicated in psoriasis pathogenesis (page 22, lines 6-7).

9) Why use two PASI scores for the binary model?

We have used PASI75 as the main binary outcome measure throughout the main figures of the study, in line with its long-held status as the gold standard for assessing treatment efficacy in moderate-to-severe psoriasis (Feldman & Krueger *Ann Rheum Dis* 2005; Puig et al. *J Eur Acad Dermatol Venereol.* 2017). Nevertheless, the availability of novel, more potent biologic therapies attaining better response as well as studies into the link between PASI values and improvement in quality of life, have generated an ongoing debate about the superiority of PASI 90 over PASI 75 (Puig et al. *J Eur Acad Dermatol Venereol.* 2015). Therefore, we also report data based on PASI90 outcome (page 10, line 1), corresponding to a global evaluation of clear or nearly clear skin, to provide a comprehensive assessment of the biomarker potential of the identified signals. Using a biomarker that can predict PASI 90 has additional value – it offers potential to identify patients who can use adalimumab – whilst putting others on to more predictably effective (albeit more expensive) biologics, such as anti-IL-17 and anti-IL-23 therapies.

10) List all 11 cell subsets

The 11 cell subsets have been specified in the main text (Page 11, Lines 14-17)

11) *The wording needs to be adjusted, it reads as the residual disease was observed in DC. “In keeping with the correlation between NF-κB nuclear translocation and residual disease observed in DCs (Figure 1d), ““Notably, however, we found a statistically significant correlation ($r^2=0.58$, $p=0.002$, $FDR < 0.05$) between LPS-induced NF-κB translocation at w0 in DCs and % residual disease, with increased NF-κB activation in patients with higher residual disease at w12”*

We are grateful to the Reviewer for highlighting the ambiguous wording in the first example above; this has now been rephrased (page 11, line 23 and page 12, line 1). However, we respectfully do not believe that the second sentence is ambiguous as it clearly distinguishes between NF-κB translocation at w0 in DCs and % residual disease.

12) *Why do you say “analytically validating”? previous assay was based on similar principles, analytically is not needed, and incorrect in this sentence*

As suggested by the Reviewer, we have removed the adverb “analytically” (Page12, line 4).

13) *“The NF-κB activation cascade includes phosphorylation of p65 NF-κB subunit, such as at Ser529, which regulate NF-κB function 17. Thus, we developed a 13-colours phospho-flow cytometry panel”, These two sentences do not have a causal link, more explanation about the need for 13 color panel is needed.*

We thank the Reviewer for flagging up this non sequitur sentence to our attention. We have revised this paragraph (page 11, line 8 and lines 10-12), explaining that to replicate and refine our previous findings, we developed a 13-colour phospho-flow cytometry panel to study NF-κB phosphorylation in 11 cell subsets within cryopreserved PBMCs of patients receiving adalimumab. The use of cryopreserved PBMCs allowed us to extend our overall sample size to 43 patients.

14) *Correlation at $r^2= 0.24$ is not really strong, it is better to confirm only LPS was predictive*

We believe that the Reviewer is referring here to the correlation between TNF-induced NF-κB phosphorylation in pDC at baseline and residual disease ($r^2=0.245$, $p=0.00509$, $FDR= 0.0433$) shown in **Figure 2e** and the inclusion of this trait in the ROC curves shown in **Supplemental Figure 15a**. We evaluated the predictive value of all 8 blood immune traits measured at w0 or w1 which displayed a statistically significant (all $FDR<0.05$) correlation with PASI75 response at w12, ranking them according to their area under the curve (AUC). First, we sought to identify the best performing biomarker for each experimental readout and ultimately the best biomarker overall. While we show the full analysis on all 8 immune traits in **Supplemental Figure 15a**, we do focus only the best performing biomarker for each readout in **Figure 6a** and perform further analyses on the overall most predictive biomarker in **Fig 6c,d,e** and **Supplemental 15b**. We respectfully consider this step-wise and comprehensive analysis to be of interest for the reader and would prefer to maintaining it as it is.

15) *It is not clear how many comparisons were done in the validation cohort – was the second one focused only in DC compartment?*

We believe that the Reviewer is referring here to the correlation between NF-κB phosphorylation and clinical response in the PSORT discovery and/or replication adalimumab

cohort. As reported in **Supplemental Table 4** we did focus only on DC subsets. This has now been clarified better in the text (page 12, lines 1, 12-14).

16) It would be easier to read if R and NR were replaced with words

As suggested, to enhance manuscript clarity R and NR have been replaced with responders and non-responders throughout the manuscript.

17) Why Th17 did not decrease? Up-regulation of CD40, CD86 and CD83 does not explain Tc17 specific stimulation

Like the Reviewer, we are also intrigued by the fact that frequency of blood Th17 does not decrease during adalimumab therapy. We also agree that up-regulation of costimulatory molecules on DC does not explain Tc17 specific stimulation, and do not suggest otherwise in the manuscript. Further studies are indeed needed to explain these observations, but we believe they fall outside of the scope of this manuscript. Nevertheless, our data are in line with the emerging role for Tc17 cells in psoriasis as their decrease in both blood and skin is needed to achieve an effective clinical response. We have expanded our discussion around Tc17 cell data (page 25, lines 19-20) to include additional considerations, also in response to *major point 3* raised by Reviewer 2.

18) “We observed a similar trend in our adalimumab cohort for CD274 protein expression in DC subsets at various time points” – it needs explaining, how did you measure a similar trend to qQTL w/o measuring SNP?

We did have genotype data for our adalimumab patient cohort (**Supplemental Figure 13b** and **Supplemental Material and Methods**, pages 1-2 Supplemental File). We have clarified this better in the main text (Page 17, line 8).

19) Discussion – why the effect is specific to DC2?

cDC2 are the major population of myeloid cDC in human blood, are high produced of IL-23 and potent activator of Th17 and CD8⁺ T cells. Moreover, cDC2 but not cDC1 were recently found to be largely increased in psoriasis lesional epidermis as compared to non lesional and healthy skin (ref 30: Reynolds et al 2021, Science). Thus, the biomarker potential of this subset is in line with their involvement on psoriasis. We have added these considerations in the revised Discussion (page 22, lines 19-23).

20) The manuscript needs editing to improve the flow, and to remove the unclear/grammatically incorrect sentences.

We are very grateful to the Reviewer for their thorough review of the text. We have carefully edited the manuscript to improve clarity and accuracy.

Reviewer #2 (Remarks to the Author):

This study represents an excellent example of deep phenotypical analysis of a psoriasis patient cohort which aims to address a significant unmet need in terms of identifying novel biomarkers which may offer utility in predicting responsiveness to adalimumab. Although the assay identified may be too cumbersome, in its present format, for routine use in the clinic, the study is novel, comprehensive and experimentally sound. It will be of significant interest to both dermatologist clinicians and researchers involved in studying the pathogenesis of psoriasis.

We thank the Reviewer for their encouraging assessment and for acknowledging the significance of our work.

With regards to the clinical utility, we note that phosphoflow cytometry, which we use to measure LPS-induced NF- κ B phosphorylation in cDC2, is emerging as a widely used tool for the discovery of biomarkers used in the diagnosis, treatment and monitoring of disease and therapy, especially in oncology (Brown et al., *Leukemia*. 2015). As already briefly reported in the Discussion (Page 26, lines 12-13), GM-CSF-induced phosphorylation of STAT5 detected by phosphoflow cytometry in peripheral blood cells has been validated to diagnose juvenile myelomonocytic leukaemia rapidly and accurately (Ref 43: Hasegawa et al., *Blood Cancer J*. 2013).

In its current form, our specific phospho-assay has the key advantage of being used in cryopreserved cells, thus allowing maximal flexibility in terms of sample collection. Moreover, there is ample scope to further develop our assay into a more clinically scalable test. Having identified cDC2 as the cell subset of interest, the panel of antibodies needed can be reduced from 13 to 8, so that the assay can be run on low-end, clinical-grade cell analyzers (e.g. BD FACSCanto II), which are widely available in clinical settings and used for clinical routine testing, rather than on high-end multiparameter cell analysers, more commonly found in research laboratories. Furthermore, it is also possible to adapt the phospho-assay to be run in whole blood (Bitar et al., *JACI*. 2017), thus removing the need to isolate and cryopreserve PBMCs, and ideally to combine the blood collection and LPS stimulation in one step, using the TrueCulture System (Duffy et al., *Clin Immunol*. 2017). Hence, we believe that, pending further development and clinical validation in prospective trials, the identified biomarker has the promise to be of clinical utility for the stratification of patients with psoriasis and potentially other TNF-driven diseases. We have concisely included these considerations in the revised manuscript (Page 26, lines 14-17).

Some specific comments include:

1. There is a discrepancy and lack of consistency in patient numbers as described in the text versus those analysed in the figures. For example, see figure 1 & 2, (16 in cohort versus 13 in 1D and 12 in 1G). If some patients have been excluded from the analysis the authors should be far more explicit for their reasons for exclusion than currently outlined in the relevant methods sections as this may have significant impact on results presented.

We are sorry for the unexplained discrepancy in patient numbers. QC criteria were pre-determined for each experiment before performing formal statistical analyses and were stated in **Material and Methods** under the respective headings. For NF- κ B nuclear

translocation, we excluded datapoints for cell type/timepoint/stimulation for which we could not acquire at least 10 cell/cell type from the overall analysis, as already previously stated in **Material and Methods** (now page 30, lines 3-4). This resulted in the apparent discrepancy highlighted by the reviewer with regards to the full discovery cohort described in **Supplemental Fig 1** (n=16), Figure 1d (n=13) and Fig 1g (n=12). For three samples it was not possible to acquire at least 10 dendritic cells in the LPS-stimulated tube at w0, hence they were not included in the analysis shown in **Fig. 1d**, and the same happened for 4 samples in the TNF-stimulated tube shown in **Fig. 1g**. For NF- κ B phosphorylation, performed on cryopreserved PBMCs, samples with cell viability <93% were not included in the overall analysis (as already previously stated, now on page 30, lines 15-16). This resulted in only 12 out of 16 discovery samples to be included in **Fig. 2b** with an overall overlap of 10 samples to be assessed for both NF- κ B translocation and phosphorylation (**Fig. 2d**). **Supplemental Table 10** originally listed the different experiments (NF- κ B nuclear translocation, NF- κ B phosphorylation etc) in which patient sample had been included. However, the table did not report samples excluded from the final analysis after QC. We have now added this information to the revised **Supplemental Table 10** by adding additional columns specifying which patient sample passed the QC for each experiment and was thus included in the final analysis. Moreover, we have re-stated the existence of QC exclusion criteria at the beginning of the revised **Statistical analysis** paragraph (page 36, line 17-19). Please see also reply to *minor point 4* raised by Reviewer 1 and *major point 5* raised by Reviewer 3.

In figure 1G, it looks as though there is a trend towards decreased Rd score in the NR patients also but this may be statistically NS due to low numbers. What is the P value here? See point about sample exclusion criteria above.

We thank the Reviewer for raising this point and have noted in the revised text the existence of a non-statistically significant ($p=0.12$) downward trend in non-responders and added p value (page 10, line 22). Please see reply above about exclusion criteria.

2. For the blood samples used in figure 1, was any analysis of relative levels of adalimumab present in patients blood undertaken post treatment undertaken? It would be interesting to determine whether individual variations in levels of circulating Ab among patients could have any impact on the observations made. This is important as this early data is framed as introducing the subsequent analysis of DC subsets.

The focus on DC subsets derives from the statistically significant correlation (**Figure 1d** and **Supplementary Table 3**) observed between clinical response at week 12 and LPS-induced NF- κ B nuclear translocation measured at baseline, that is before patients commence adalimumab therapy, i.e. in absence of the drug. Nevertheless, we agree with the Reviewer that adalimumab levels in the blood of the patients could have had an impact on the post-baseline results of the correlation analysis shown in **Supplementary Table 3**. To address this point, we carried out linear regressions of relative PASI on NF- κ B translocation at w1, 4 and 12 and drug concentration simultaneously. The results of this new analysis, shown in a new **Supplementary Table 4**, confirms that clinical response is not correlated with NF- κ B nuclear translocation in any combination of stimulation, cell type and time point, even after accounting for variation of drug concentration between patients ($FDR > 0.05$). We report this

new findings in the manuscript on page 10, lines 12-17 and describe methodology on page 30, lines 13-16 and statistical analysis on page 37, line 8-11.

3. The observations concerning the relative levels of Tc17 cells and their possible predictive role in determining anti-TNF responsiveness are intriguing. However data from the ROC curve analysis in Supp. Fig 15 indicate that Tc17 levels at week 0 are not as good accurate predictor of response. These data should be discussed in more detail.

Like the Reviewer, our interest was also piqued by the Tc17 cell data in blood and skin, especially when compared and contrasted with Th17 cell data. We detected an overall significant decrease of Tc17 cells in blood and skin during adalimumab treatment and found a small, albeit statistically significant, positive correlation between their baseline level and lack of clinical response at w12. Of note, we detected a stronger and more significant correlation with response for baseline levels of blood Th17 cells and w12 Tc17. Interestingly, however, Tc17, but not Th17 cells remained elevated in the skin of adalimumab non responder at w12. Based on these findings and on the results of the ROC analyses in **Supplemental Fig 15a** we speculated in Discussion (page 25, lines 19-20) that while baseline levels of blood IL-17+ T cells, particularly Th17, could have a role in predicting clinical outcome, a significant decrease of Tc17 cells in both blood and skin is needed to achieve an effective clinical response. Thus, Tc17 cells may potentially play a role as dynamic or surrogate biomarker, i.e. a biomarker that can be used to monitor the effect of the drug or to substitute for a clinical endpoint (as defined by Robinson et al. 2013, Nat Rev Rheum, ref. 11), rather than a predictive biomarker. While there is no clinical need for a surrogate biomarker at w12 in psoriasis, these observations provide further insights into disease pathogenesis and adalimumab mechanism of action. Nevertheless, the stronger predictive value of LPS-induced NF- κ B p65 phosphorylation in cDC2 strongly points to clinical response to adalimumab more likely to be directly determined, as well as predicted by the intrinsic properties of DC. We have expanded our discussion around Tc17 cell data to include the above considerations on page 25, line 23 and page 26, lines 1-2.

Minor comments:

1) Typos throughout the manuscript e.g. line 220, line 538

We thank the reviewer for their thorough review of the text. We have corrected typos throughout.

2) Representative FACS plots in Figs 1A & 2A should be labelled

Panels have been labelled as requested. Please see also reply to minor point 1 raised by **Reviewer 1**.

Reviewer #3 (Remarks to the Author):

Andres-Ejarque and colleagues report that the activation potential of dendritic cells, especially cDC2 cells, at baseline correlates with the clinical outcome of adalimumab therapy at week 12 in psoriasis patients. The study is part of a large consortium aiming at identifying psoriasis biomarkers. Such biomarkers of response are a high unmet medical need. The manuscript is well written and of high interest to the broad readership of Nature Communications. The complex nature of the biomarker

somewhat dampens the enthusiasm of clinical utility, but the degree of innovation is very high and the manuscript should be published after some corrections.

We thank Reviewer 3 for their positive assessment of our work and address their points below.

With regards to the clinical utility, we note that phosphoflow cytometry, which we use to measure LPS-induced NF- κ B phosphorylation in cDC2, is emerging as a widely used tool for the discovery of biomarkers used in the diagnosis, treatment and monitoring of disease and therapy, especially in oncology (Brown et al., *Leukemia*. 2015). As already reported in Discussion (Page 26, lines 12-14) phosphorylation of STAT5 detected by phosphoflow cytometry in peripheral blood cells in response to GM-CSF has been validated to diagnose juvenile myelomonocytic leukaemia rapidly and accurately (Ref 43: Hasegawa et al., *Blood Cancer J*. 2013).

In its current form, our specific phospho-assay has already the key advantage to be performed in cryopreserved cells, thus allowing maximal flexibility in terms of sample collection. Moreover, there is ample scope to further develop it into a more clinically scalable test. Having identified cDC2 as the cell subset of interest, the panel of antibodies needed can be reduced from 13 to 8, so that the assay can be run on low-end, clinical-grade cell analyzers (e.g. BD FACSCanto II), which are widely available in clinical settings and used for clinical routine testing, rather than on high-end multiparameter cell analysers, more commonly found in research laboratories. Furthermore, it is also possible to adapt the phospho-assay to be run in whole blood (Bitar et al., *JACI*. 2017), thus removing the need to isolate and cryopreserve PBMCs, and ideally to combine the blood collection and LPS stimulation in one step, using the TrueCulture System (Duffy et al., *Clin Immunol*. 2017). Hence, we believe that the identified biomarker has a realistic potential to be of effective clinical utility. We have concisely included these considerations in the revised manuscript (Page 26, lines 14-17).

1.) Statistical analysis: the ratio of n number in experiments and number of readouts requires a Bonferroni-correction. How do corrected p values look like? There is a risk of over-interpretation.

We agree with Reviewer 2 that correction for multiple testing is necessary and in fact had already applied False Discovery rate (FDR) correction to all correlation analyses between a cellular readout (i.e. NF- κ B nuclear translocation in immune cells; NF- κ B phosphorylation in DC subsets; frequency of cytokine-producing T cell subsets; phenotypic marker expression in DC subsets) and % residual disease, as reported in related Figures and Supplemental tables, e.g. **Figure 1d** and **Supplemental Table 2** for NF- κ B translocation. We have chosen to apply FDR, instead of the more conservative Bonferroni, based on our methodological approach to perform a series of follow-up experiments to replicate and validate the initial findings. Bonferroni corrections control for the family-wise error rate, i.e. the probability of having one or more false positives out of all the hypothesis tests conducted. While this approach guards against the occurrence of false positives, it may lead to a considerable number of missed findings. FDR, instead, is a way to identify as many significant features as possible, while incurring a relatively low proportion of false positives and is particularly useful when initial findings are prioritised for further validation, as in our study.

Moreover, while multiple correction testing is often not applied at all in deep phenotyping studies, FDR appears to be indeed the method of choice (Hartmann et al., *Cell Reports*. 2019; Nowicka et al., *F1000Res*. 2017).

Nevertheless, to address any specific concern the Reviewer may have about the key finding of the study, i.e. the correlation between LPS-induced NF-κB phosphorylation at baseline in cDC2 and residual disease, we have rerun the analysis applying Bonferroni correction and the result is still statistically significant ($p_{adj} = 7.43 \times 10^{-5}$) (data for the Reviewer only, not included in the revised manuscript).

2.) *Influence of skin severity – many findings in the manuscript point towards the possibility that non-responders had a more severe inflammation at baseline than responders, and this is reflected by the activation potential of DCs. In other words, does the level of NF-κB translocation merely reflect disease severity and the more severely affected patients respond less well to adalimumab? Also results described in lines 247/248 and the higher number of IL-17+ cells in non-responders shown in figure 5 would argue in that direction. The authors address that in supplemental table 5, which is not very well explained, and only for the cDC2 experiments in figure 2. It would help to clarify here.*

We thank the Reviewer for this important comment. We have acknowledged in the text (page 5, line 1) that disease severity, measured as baseline PASI, has been previously identified, among others already cited, as clinical predictor of outcome to biologics (Warren et al., *Br J Dermatol.* 2019; ref. 8 in the manuscript). Indeed we had already taken this into account in our analysis, as we had included baseline PASI as a covariate (alongside age, gender, ethnicity, smoking, weight, psoriatic arthritis, being biologic naïve, or the presence of the HLA-C*06:02 allele), for all 3 statistically significant correlations described in lines 247/248 of the original submission (LPS-induced phospho NF-κB in cDC2, TNF-induced phospho NF-κB in cDC2 and pDC at baseline, $FDR \leq 0.05$), as shown in **Supplemental table 5**. Baseline PASI did not have a significant effect for any of the cell types and stimulations tested, including our lead signal in LPS-induced phospho NF-κB in cDC2, showing that the correlation between NF-κB activation and lack of clinical response is independent from disease severity. We described the methodology applied in **Supplemental Methods** under *Covariate Analysis* (page 1, Supplemental File, as in the original submission). Moreover, we have now added the median baseline PASI for PASI75 responders and non responders included in the Combined adalimumab cohort (n=43) in the revised **Materials and Methods**, under *Study design and patient cohort* (page 27, lines 18-19). Finally, to address the specific query about the higher number of IL-17+ cells in non-responders shown in **Figure 5f**, we now report that the median PASI at week 0 for responders (12.9 ± 3.1) and non-responders (13.0 ± 4.7) were comparable in the two groups of selected PASI75 responders and non responders (n=20) used for the immunostaining experiments, thus confirming that the difference observed are not driven by disease severity (page 18, lines 9-10). Finally, we now mention the lack of effect of covariates, like disease severity on our findings in the discussion (page 21, line 6; page 25, line 12)

3) *Overinterpretation: results table 2 and figure 1c: please modify sentence lines 195/196, as we would always expect a delayed clinical response to molecular changes at cellular level.*

We thank the reviewer for raising this point as this sentence and the previous one were in fact not clear and may have generated some confusion, i.e. that we expected to see a clinical response at the same time as the molecular changes at cellular level were taking place. Instead, we always assess clinical response at w12. We have now edited these two sentences: i) clarifying that no correlation between TNF-induced NF-κB nuclear translocation *at any time*

point and % residual disease at w12 was detected in any cell type (page 10, lines 2-3), despite the inhibitory effect of adalimumab in lymphoid cells; *ii*) *suggesting* that complete inhibition of NF- κ B signalling in T, NK and NKT cells is not likely to underpin clinical response to adalimumab at w12 (page 10, line 45).

Similarly, Lines 299/300 – why do the authors refer to cDC2 if moDC were investigated?

We thank the Reviewer for highlighting this inaccuracy. We have modified cDC2 into DC (page 15, line 4) to link together more correctly the in vitro moDC experiments with the previously shown ex vivo experiments.

Finally, the statement in lines 304-306 is not supported by data – there is no mechanistic link established in the work between NF-kB activation potential of DC subsets and the number of IL-17+ T cells.

We agree with the Reviewer that the data do not establish a mechanistic link between the two events, and simply report their simultaneous occurrence. Therefore, we have rephrased the sentence, removing removed the reference to a mechanistic link (page 21, lines 8-13).

4) Statistics: Line 412/413: was the cut-off defined before the validation cohort was analysed? If not, there is a risk of overfitting

The cut-off was calculated in the PSORT combined cohort (**Fig. 6b**, n= 28 patients) and was then applied to the independent clinical validation cohort (Fig 6e, n=15) to assess its predictive value. We have clarified this better in the text (Page 20, line 8) and in the legend to **Figure 6e** (page 50, line 17)

5) Figure 1: relative PASI calculation, the remaining PASI – is it true that all patients improved by at least 50% after week 12 under Adalimumab, and how could the different picture in the discovery cohort (figure 2) influence the findings?

The Reviewer refers here to the apparent discrepancy between the relative PASI of the discovery cohort shown in **Fig. 1d**, where all 13 patients improved by at least 50%, and in **Fig. 2b**, where 3 of the 13 patients depicted had less than 50% improvement. Although both panels are based on the PSORT discovery cohort (comprising 16 patients in total), the individual data points included in each analysis may refer to different patients, based on the results of the quality control applied to ensure robustness of the data. When assessing NF- κ B nuclear translocation by imaging flow cytometry in fresh blood, we excluded from the overall analysis datapoints for cell type/timepoint/stimulation for which we could not acquire at least 10 cell/cell type, as already previously stated in **Material and Methods** (now on page 30, lines 3-4). When assessing NF- κ B phosphorylation in cryopreserved PBMC, only cells with $\geq 93\%$ post-thawing viability were included in the analysis, with the viability-cut-off determined as already previously described in **Material and Methods** (now on page 31, lines 15-16) and shown in **Supplemental Figure 16**. Consequently, some patient samples passed the QC filter for one experiment but not for the other. The patient overlap between the two experiments is shown in **Fig 2d**, showing that we have measured both NF- κ B nuclear translocation and phosphorylation in the same 10 patients. In the revised **Material and Methods**, we have restated the existence of data exclusion criteria specific for each experiment and that these may

give rise to apparent discrepancies in patient numbers under the Statistical analysis paragraph (page 36, line 17-19). Moreover, we now specify in **Supplemental table 10** which patients are included in the analysis for each experiment. Please see also reply to *minor point 4* raised by reviewer 1 and *major point 1* raised by Reviewer 2.

REVIEWERS' COMMENTS

Reviewer #1 (Remarks to the Author):

The manuscript clarity has greatly improved on the revision, and provides an insightful and compelling reading.

All my queries has been answered in sufficient details.

Only minor points are outstanding, suggestions given on how to improve clarity:

- p10, lines 6-9: The sentence: "Notably, however, we found a statistically significant correlation ($r^2=0.58$, $p=0.002$, $FDR\ 6 < 0.05$) between LPS-induced NF- κ B translocation at w0 in DCs and % residual disease, with increased NF- κ B activation in patients with higher residual disease at w12 (Figure 1d and 8 Supplemental Table 3)." is difficult to follow. It feels like too many correlations are reported at once, (correlation between LPS induced NfKB and % disease with increased NFkB activation...) I would suggest splitting into two separate sentences for each claim.

- p10, line 19: "patients achieving PASI75 and 18 PASI90 response" - figure 1F is cited, while it should refer to Figure 1G

- p13 line 8 "was at the same level than that observed in HV" -> the same level as

-p16 line 10 onwards - this section reports changes in DC, not in Th17, I think it deserves a separate title, to differentiate from T cell results

- Figure 4c, bottom panel - no of weeks not stated

- p18 line 12 : "as well as the number 11 of IL23+ ($p<0.01$) DC at week 12" I suggest adding "in all patients/in both responders and non-responders" to improve clarity

-p18 line 13 "Moreover, PASI75-12 non-responders harboured increased number ($p<0.05$) of CD11c+ DC" I suggest using "However" instead of "Moreover" as the finding is differentiating responders vs non, in contrast to the previous one showing a change in complete cohort

With best wishes

Marta E Polak

Reviewer #2 (Remarks to the Author):

All of my comments have been adequately addressed.

Reviewer #3 (Remarks to the Author):

The authors have revised the manuscript for the sake of clarity, now avoid unnecessary speculations and have explained methodological decisions taken to my satisfaction. This manuscript remains a highly valuable contribution to the community. Further studies have to evaluate the clinical utility of the proposed biomarker. Congratulations! Kilian Eyerich

REVIEWERS' COMMENTS

Reviewer #1 (Remarks to the Author):

The manuscript clarity has greatly improved on the revision, and provides an insightful and compelling reading. All my queries has been answered in sufficient details.

We thank Dr Polak for her positive remarks and constructive feedback.

Only minor points are outstanding, suggestions given on how to improve clarity:

- p10, lines 6-9: The sentence: "Notably, however, we found a statistically significant correlation ($r^2=0.58$, $p=0.002$, $FDR < 0.05$) between LPS-induced NF- κ B translocation at w0 in DCs and % residual disease, with increased NF- κ B activation in patients with higher residual disease at w12 (Figure 1d and 8 Supplemental Table 3)." is difficult to follow. It feels like too many correlations are reported at once, (correlation between LPS induced NfKB and % disease with increased NfKB activation...) I would suggest splitting into two separate sentences for each claim.

We have now split the sentence as suggested to improve clarity. The revised text reads: "However, we found a statistically significant correlation ($r^2=0.58$, $p=0.002$, $FDR < 0.05$) between LPS-induced NF- κ B translocation at w0 in DCs and % residual disease, (Figure 1d and Supplemental Table 3). In particular, patients with higher residual disease at w12 displayed increased NF- κ B activation at baseline".

- p10, line 19: "patients achieving PASI75 and 18 PASI90 response" - figure 1F is cited, while it should refer to Figure 1G

We respectfully note that here we are indeed referring to Figure 1F, showing decreased LPS-induced NF- κ B translocation at w0 in adalimumab patients achieving PASI75. Figure 1g refers to TNF-induced NF- κ B activation.

- p13 line 8 "was at the same level than that observed in HV" -> the same level as

This has been changed as suggested

-p16 line 10 onwards - this section reports changes in DC, not in Th17, I think it deserves a separate title, to differentiate from T cell results

Thank you for the suggestion, we have added the following title: "Response to adalimumab is associated with early upregulation of CD274 in DCs"

- Figure 4c, bottom panel - no of weeks not stated

We respectfully note that weeks are listed underneath the cell type, e.g "%Tc17 w12".

- p18 line 12 : "as well as the number 11 of IL23+ ($p<0.01$) DC at week 12" I suggest adding "in all patients/in both responders and non-responders" to improve clarity

"In all patients" has been added as suggested.

-p18 line 13 "Moreover, PASI75-12 non-responders harboured increased number ($p<0.05$) of CD11c+

DC" I suggest using "However" instead of "Moreover" as the finding is differentiating responders vs non, in contrast to the previous one showing a change in complete cohort

This has been changed as suggested

With best wishes
Marta E Polak

Reviewer #2 (Remarks to the Author):

All of my comments have been adequately addressed.

We are pleased to have met Reviewer's 2 requests and thank them for their useful feedback.

Reviewer #3 (Remarks to the Author):

The authors have revised the manuscript for the sake of clarity, now avoid unnecessary speculations and have explained methodological decisions taken to my satisfaction. This manuscript remains a highly valuable contribution to the community. Further studies have to evaluate the clinical utility of the proposed biomarker. Congratulations! Kilian Eyerich

We thank Prof Eyerich for his helpful feedback and supportive remarks.